# Revisiting Semi-Supervised Learning in the Era of Foundation Models

**Ping Zhang**[*][†]   **Zheda Mai**[*]   **Quang-Huy Nguyen**   **Wei-Lun Chao**
The Ohio State University
{zhang.14217, mai.145, nguyen.2959, chao.209}@osu.edu

## Abstract

Semi-supervised learning (SSL) enhances model performance by leveraging abundant unlabeled data alongside limited labeled data. As vision foundation models (VFMs) become central to modern vision applications, this paper revisits SSL in the context of these powerful pre-trained models. We conduct a systematic study on tasks where frozen VFMs underperform and reveal several key insights when fine-tuning them. First, parameter-efficient fine-tuning (PEFT) using only labeled data often surpasses traditional SSL methods—even without access to unlabeled data. Second, pseudo-labels generated by PEFT models offer valuable supervisory signals for unlabeled data, and different PEFT techniques yield complementary pseudo-labels. These findings motivate a simple yet effective SSL baseline for the VFM era: *ensemble pseudo-labeling across diverse PEFT methods and VFM backbones*. Extensive experiments validate the effectiveness of this approach, offering actionable insights into SSL with VFMs and paving the way for more scalable and robust semi-supervised learning in the foundation model era.

## 1   Introduction

The quality of machine learning (ML) models is often closely tied to the amount of labeled data available, but annotation can be costly or labor-intensive. **Semi-supervised learning (SSL)**, which leverages abundant unlabeled data alongside limited labeled data, has thus emerged as a promising paradigm for developing ML models without the need for extensive labeling [82, 10, 67, 74]. Over the past few decades, numerous SSL algorithms have been developed to advance this field. Exemplar methods from the deep learning era include Mean Teacher [61], MixMatch [6], and FixMatch [59], which dynamically impose objective functions on unlabeled data based on distillation and consistency, thereby enhancing learning performance. *It is worth noting that many of these methods were originally designed to train neural networks "from scratch."*

Recently, pre-training on external labeled or unlabeled data has become the de facto standard across many machine learning application domains [29, 46]. For example, in computer vision, many recent algorithms are built upon **vision foundation models (VFMs)** such as CLIP [55] and DINOv2 [53]. These models were pre-trained on massive datasets—millions, if not billions, of data points. As a result, they have demonstrated remarkable generalizability across a wide range of tasks, often requiring only slight fine-tuning or, in some cases, functioning effectively as a frozen backbone [79].

Given the promising advancements in both fields, we explore their interaction in this paper. Specifically, we seek to address the following questions: Are existing SSL algorithms still effective when using VFMs as the backbone? What adjustments, if any, are needed to improve their performance? Finally, can we leverage the power of VFMs to develop more effective, yet simpler, SSL algorithms?

---

[1]Equal contribution.

[2]Corresponding author: zhang.14217@osu.edu.

 Our code is available at `https://github.com/OSU-MLB/SSL-Foundation-Models`.

39th Conference on Neural Information Processing Systems (NeurIPS 2025).

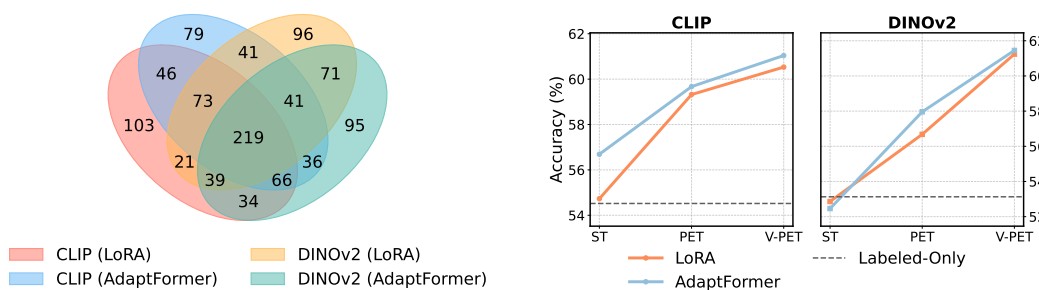

Figure 1: *Left*: The Venn diagram of the top 20% highest-confidence predictions from various VFMs and PEFT under DTD 3-shot, illustrating their intrinsic property of producing a diverse range of high-confidence predictions. *Right*: Ensembling these diverse pseudo-label predictions progressively boosts downstream performance for increasingly more ensembles (Self-Training → PET → V-PET), highlighting the quality improvements from diversity. Results are averaged across 12 settings in our benchmark.

**Study design.** To this end, we introduce new SSL benchmark datasets based on the Visual Task Adaptation Benchmark (VTAB) [76], a diverse suite of classification tasks designed to evaluate visual representations. *Our focus is on tasks where frozen VFMs underperform, indicating the need for further fine-tuning, and where SSL could offer a beneficial solution.* We then systematically evaluate three representative SSL methods—FixMatch [59], FlexMatch [77] and SoftMatch [11]. Hyperparameters are carefully selected using techniques proposed for unsupervised domain adaptation [23], ensuring that the data leakage issue discussed in [52] is avoided.

**Key insights.** Our empirical results highlight two main findings. First, fine-tuning VFMs with representative SSL algorithms offers limited advantages over using only labeled data for fine-tuning. Second, **parameter-efficient fine-tuning (PEFT)** [47, 72, 60]—which updates a small subset of parameters or adds lightweight learnable modules while keeping the VFM largely frozen—consistently yields substantial performance gains, regardless of the learning paradigm.

These observations imply two important directions. First, there is a need for SSL methods tailored specifically for VFMs. Second, since PEFT models trained only on labeled data already match the performance of standard SSL approaches, effectively leveraging their predictions—*i.e.*, pseudo-labels—offers a promising path to further improve performance in semi-supervised settings.

**An SSL baseline in the VFM era.** Building on these insights, we introduce a simple yet effective SSL approach that leverages VFMs as backbones. Our method is grounded in the principle of **self-training**—a straightforward SSL strategy in which the model generates pseudo-labels for unlabeled data to guide further training [41, 71, 13, 73]. While traditional self-training often struggles with low-quality pseudo-labels, we address this by exploiting two key properties of VFMs and PEFT methods: their strong initial performance and complementary behaviors.

Specifically, as observed in [47, 63], different VFM backbones and PEFT methods frequently produce diverse predictions—even when their overall accuracies are similar. As shown in Figure 1, this diversity motivates **ensembling** predictions from multiple VFM-PEFT pairs [81]. By explicitly compensating for their varied confidence distributions, we obtain significantly more robust pseudo-labels—*without requiring explicit filtering* [59, 51]. The result is a simpler yet more reliable self-training pipeline (Figure 2) that effectively leverages both labeled and unlabeled data, achieving substantial gains over existing SSL methods.

We extensively validate our approach, **V**FM-**P**EFT **E**nsemble **T**raining (**V-PET**), on newly proposed benchmark datasets. V-PET consistently outperforms existing SSL methods across most tasks, including the recently proposed FineSSL [25], which also builds on VFMs. These results establish PET as a simple, effective, and competitive SSL baseline for the foundation model era.

**Remark.** Self-training [1, 2], pseudo-labeling [41], and ensemble methods [20] have been extensively studied in the ML literature, spanning both the pre-deep learning and deep learning eras. Our goal is

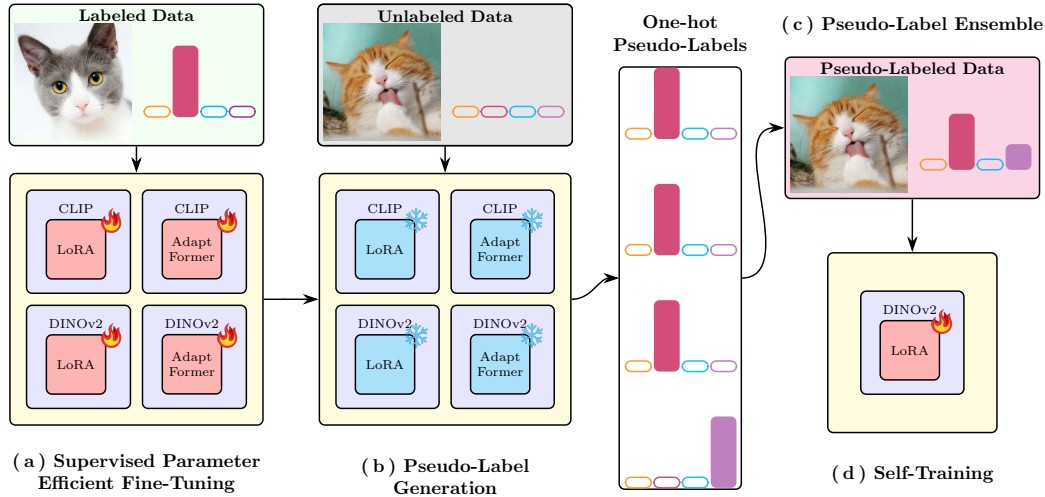

Figure 2: Illustration of V-PET. To effectively leverage abundant unlabeled data alongside scarce labeled data in the era of VFMs, our approach follows four phases: (a) *Supervised Parameter-Efficient Fine-Tuning*, where we harness labeled data by fine-tuning pre-trained VFMs using various PEFT algorithms; (b) *Pseudo-Label Generation*, where we exploit fine-tuned VFMs' generalization ability to generate pseudo-labels for unlabeled data; (c) *Pseudo-Label Ensemble*, where we enhance robustness by aggregating pseudo-labels from multiple fine-tuned VFMs; and (d) *Self-Training*, where we consolidate all knowledge into one model.

not to compete with existing methods but to *establish a simple yet effective semi-supervised learning baseline* that harnesses their strengths while incorporating the unique properties of foundation models. We note that a key step in ensembling is obtaining multiple base learners that are diverse and equally performant. Our contribution lies in leveraging the complementary behaviors of multiple foundation models and PEFT methods—an approach tailored specifically for the foundation model era.

Consistent with representative SSL studies [6, 59, 77], our primary focus is on classification. Nonetheless, we believe that our insights are transferable to various tasks and hope they will inspire future research on other downstream SSL tasks, such as segmentation and detection.

## 2  Related Works

**Semi-Supervised Learning (SSL).** In recent years, many SSL methods have centered on generating and selecting reliable pseudo-labels [59, 6, 5]. FixMatch [59] uses a fixed confidence threshold, while FlexMatch [77] adopts class-specific thresholds and SoftMatch [11] applies a soft threshold to balance label quality and quantity. However, these approaches were designed for training from scratch, leaving open questions about their compatibility with VFMs. Another key paradigm is self-training [71, 41], which (1) trains a teacher on labeled data, (2) generates pseudo-labels for unlabeled samples, and (3) trains a student on both. Ensuring pseudo-label reliability typically involves unsupervised pre-training [13], confidence thresholds, or consistency constraints [59, 51]. We instead leverage VFMs to produce more robust pseudo-labels, questioning whether scratch-oriented SSL algorithms remain effective in the foundation-model era.

**Transfer Learning & Self-Supervised Learning.** Transfer learning [83, 65]—and specifically PEFT (or PETL) for efficiently adapting foundation models [31]—has long leveraged pretrained models to boost downstream tasks. In our approach, we fine-tune VFMs via PEFT on labeled data, then use the adapted models to generate pseudo-labels for unlabeled examples, thus bridging transfer learning with semi-supervised methods. A related paradigm, self-supervised learning, also exploits unlabeled data and often serves as a baseline in SSL comparisons [13, 2], but it assumes abundant unlabeled data—an assumption that may fail in low-resource settings.

**Vision Foundation Models.** Vision Transformers (ViT) [21] pre-trained on massive amounts of data, have become indispensable to modern AI development [3, 45, 28]. These models, often referred to as vision foundation models (VFMs), have demonstrated superior performance on a wide range of

tasks. For example, CLIP-ViT [55] trained with millions of image-text pairs shows an unprecedented zero-shot capability, robustness to distribution shifts, and serves as the encoders for various powerful generative models [57, 42]. Meanwhile, DINOv2 [53] pre-trained with self-supervised objectives on extensive sets of well-curated images effectively captures fine-grained localization features. Moreover, it is increasingly becoming common sense that strategically fusing multiple VFMs can unlock synergistic gains [47, 63], leading to state-of-the-art performance across tasks from visual question answering to object detection.

**Leveraging VFMs for Semi-Supervised Learning.** With the growing prominence of VFMs, few recent studies have explored their usage in SSL. The most relevant to ours is FineSSL [25], which specifically investigates the use of CLIP vision backbone in an SSL setting, accompanied by pseudo-labels refinement using a balanced margin softmax. The evaluation was limited to simple, small-scale datasets like CIFAR-10 [40], where frozen VFMs can already achieve high accuracy. A few other works constrained the study only to VFMs with zero-shot capabilities, such as the full CLIP model with both vision and text encoders [50, 78]. Our work, in contrast, extends the scope in both the diversity of VFMs and the evaluation benchmarks, establishing the first comprehensive study of SSL in the era of foundation models.

Table 1: Comparison of supervised linear probing performance (%) between popular SSL datasets (CIFAR, Food101) and our benchmark, where N denotes the number of labeled samples per class. While frozen VFMs already excel in standard SSL datasets, they struggle in our benchmark—comprising diverse tasks, domains, and sizes—underscoring the potential of SSL for unleashing the full potential of VFMs.

| | CIFAR-10 | CIFAR-100 | FOOD-101 | DTD | SUN397 | RESISC45 | Retinopathy | CLEVR-C | KITTI |
|---|---|---|---|---|---|---|---|---|---|
| N | 4 | 100 | 10 | 6 | 6 | 2 | 80 | 20 | 10 |
| CLIP | 85.0 | 78.3 | 80.2 | 61.8 | 63.7 | 69.3 | 35.9 | 33.1 | 51.1 |
| DINOv2 | 91.7 | 88.1 | 83.1 | 66.7 | 65.3 | 52.4 | 41.2 | 30.5 | 51.3 |

# 3   On Evaluation of SSL in the Era of VFMs

## 3.1   Problem Definition

We consider a $C$-class classification problem in SSL with a large unlabeled dataset $\mathcal{U} := \{x_i^u\}_{i=1}^{|\mathcal{U}|}$ and a much smaller labeled dataset $\mathcal{L} := \{(x_i^l, y_i^l)\}_{i=1}^{|\mathcal{L}|}$, where $x_i^l$ and $x_i^u$ are training samples and $y_i^l$ is the ground-truth class label. $|\mathcal{U}|$ and $|\mathcal{L}|$ denote the unlabeled and labeled dataset size with $|\mathcal{U}| \gg |\mathcal{L}|$. SSL aims to learn a model $f_\theta$ parameterized by $\theta$ using $\mathcal{U}$ and $\mathcal{L}$. Unlike conventional SSL that initializes $\theta$ randomly, our framework starts with a VFM (*e.g.*, , CLIP or DINOv2) and fine-tunes it for downstream tasks.

## 3.2   A Comprehensive SSL Image Classification Benchmark

Despite recent advancements in SSL, most studies [25, 11, 77] continue to evaluate on classic datasets such as CIFAR-10/100 [40], STL-10 [17], and Food101 [8]. However, these benchmarks exhibit two key limitations: **Diminishing difficulty under VFMs.** Following prior PEFT-on-VFM work [37, 35, 66, 47, 70, 80], we use linear probing as our primary performance indicator. As shown in Table 1, linear probing on frozen VFM backbones already delivers remarkable accuracy—even with limited labeled data. We also report results for various classification-head sizes in appendix B.4, and **Narrow domain coverage.** Because these benchmarks focus mainly on natural images, they offer only a narrow view of real-world VFM applications. To effectively evaluate SSL methods in the VFM era—focusing exclusively on semi-supervised image classification—we propose a new benchmark designed to capture the complexities of real-world applications across different domains and dataset sizes.

**Dataset & Regime** Following the VTAB protocol [76], we select six classification datasets covering the three VTAB categories— *Natural* , *Specialized* , and *Structured* . Specifically, we choose two datasets from each category: DTD and SUN397 from *Natural* , RESISC45 and Retinopathy from *Specialized* , and CLEVR-C and KITTI from *Structured* . These datasets span diverse domains, including texture recognition, scene understanding, remote sensing, medical imaging, synthetic reasoning, and autonomous driving. To evaluate the robustness of SSL

methods, we vary the number of labeled samples per class and adopt *linear probing* as the evaluation protocol, with shot counts chosen to keep each task sufficiently challenging for frozen VFMs. This configuration imposes substantial difficulty on frozen representations (Table 1), underscoring the necessity of SSL for unlocking their full potential. A summary of the datasets appears in Table 2, and detailed descriptions are provided in Appendix A. Together, this benchmark offers a more diverse and comprehensive foundation-model-era evaluation of semi-supervised image classification.

### 3.3  Fair Hyperparameter Tuning for SSL

Hyperparameter tuning has long been a persistent challenge in SSL [52], remaining ambiguous and unstandardized throughout much of the existing literature. Due to the labeled data scarcity, a standard train-validation split is often infeasible. Tuning on a held-out labeled validation or even test set can lead to significant data leakage, resulting in over-optimistic results and unfair comparisons. To tackle this issue and complement our benchmark, we establish a more rigorous protocol for hyperparameter tuning in SSL.

Table 2: Summary of our benchmark. Rem. Sen.: Remote sensing; Recog.: Recognition; $|\mathcal{L}|$: # labeled training data; $|\mathcal{U}|$: # unlabeled training data.

| Dataset | Task | Domain | $|\mathcal{L}| + |\mathcal{U}|$ | Classes | $|\mathcal{L}|$/Class |
|---|---|---|---|---|---|
| DTD | Recog. | Textural | 3,008 | 47 | 3, 6 |
| SUN397 | Recog. | Natural | 49,601 | 397 | 3, 6 |
| RESISC45 | Recog. | Rem. Sen. | 20,160 | 45 | 1, 2 |
| Retinopathy | Recog. | Medical | 36,825 | 5 | 4, 8 |
| CLEVR-C | Count | Synthetic | 56,000 | 8 | 1, 2 |
| KITTI | Depth | Auto Drive | 5,416 | 4 | 5, 10 |

Our core insight is to harness the defining characteristic of SSL—an abundance of unlabeled *training* data—to tune parameters in an unsupervised manner and avoid the pitfalls of data leakage. One recent study has explored such an idea for domain adaptation, using unsupervised criteria such as RankMe [26] and AMI [68] to estimate the effectiveness of each hyperparameter configuration [23].

However, [23] also revealed that no single criterion could reliably select suitable hyperparameters across all scenarios. Motivated by this finding, we propose integrating **seven** unsupervised criteria—five derived from features, AMI [68], ARI [30], V-Measure [58], FMI [24] and BNM [18], and two from logits, RankMe [26] and CHI [9]—for more robust tuning. Concretely, for each hyperparameter configuration and its corresponding model, we compute all seven criteria using a held-out *unlabeled* validation set $\mathcal{V} := \{\boldsymbol{x}_i^v\}_{i=1}^{|\mathcal{V}|}$. Next, we rank every hyperparameter configuration based on each criterion and choose the one achieving the lowest average rank across all. We provide formal definitions and detailed procedures in appendix C.1 and the effectiveness of the proposed method in section 6.3. By eliminating reliance on held-out labeled validation sets, our method mitigates data leakage and promotes a more practical and fair tuning protocol for SSL.

## 4  Systematic Evaluation of SSL with VFMs

Given the struggling performance of frozen VFMs in our benchmark (Table 1), we consider two straightforward strategies to enhance their effectiveness: (1) exploit the inherent generalizability of VFMs by fine-tuning only on labeled data; (2) employ SSL to leverage both labeled and unlabeled data. With the diverse benchmark and hyperparameter tuning protocol introduced in section 3, we aim to investigate whether existing SSL algorithms remain effective when adopting VFMs as their backbone.

**Evaluation Setup.** We focus on two representative VFMs—ViT-B/16 CLIP [55] and ViT-B/14 DINOv2 [53]—covering language-image contrastive pre-training and self-supervised pre-training strategies, respectively. We examine four representative SSL methods, including FixMatch [59], FlexMatch [77], SoftMatch [11], and FineSSL [25] and use labeled-only fine-tuning as the baseline. We employ the AdamW optimizer [44] with a batch size of 32 and weight decay $5 \times 10^{-4}$ to fine-tune the model for 35 epochs. Learning rates and other hyperparameters are tuned with our proposed tuning protocol. The classification performance is evaluated using the Top-1 accuracy on the test dataset.

**Surprising Effectiveness of Labeled-only Fine-tuning.** Surprisingly, under fair comparison, *full* fine-tuning with even a few labeled images per class can match or surpass SSL methods, as illustrated in Figure 3. In other words, even with a large amount of additional unlabeled data, SSL provides little advantage over fine-tuning VFMs with limited labeled data. This finding is reminiscent of observations made in a comprehensive investigation conducted seven years ago [52].

Due to the inherently noisy supervised signals associated with unlabeled data in SSL methods, we hypothesize that allowing SSL to update all parameters in VFMs may inadvertently reduce their built-in generalizability.

**Can PEFT Come to the Rescue?** When labeled downstream data are scarce, recent research has shown that parameter-efficient fine-tuning (PEFT)—which updates only a small subset of parameters or introduces a lightweight learnable module to frozen VFMs—often outperforms full fine-tuning on VFMs [47, 72, 32, 12, 35, 66]. When considering SSL scenarios with limited labeled data, a natural question arises: is PEFT compatible with SSL? Although a recent study has made an initial attempt to apply PEFT in SSL [25], a comprehensive analysis of its compatibility with different VFM backbones, PEFT methods, and SSL approaches is still lacking. We, therefore, assess two commonly used PEFT strategies: LoRA [32], which trains additive low-rank matrices for Transformer layers to approximate weight updates, and AdaptFormer [12], an adapter-based approach proven effective in computer vision. We follow the same protocol outlined in section 3 to tune the hyperparameters of these PEFT methods.

Our results in Figure 3 show that PEFT indeed improves SSL on VFMs. However, it also enhances labeled-only fine-tuning, with results remaining comparable to SSL. This highlights the limited effectiveness of using unlabeled data in existing SSL methods when paired with VFMs.

**Discussion.** Three key takeaways emerge from these findings. First, a well-tuned labeled-only PEFT serves as a competitive baseline for SSL with VFMs. Second, the ineffectual use of unlabeled data in existing SSL methods underscores the need for SSL approaches specifically designed for VFMs. Finally, given that labeled-only PEFT can achieve accuracy comparable to SSL, its predictions on unlabeled data generate pseudo-labels of sufficient quality. A promising direction is to leverage these pseudo-labels to further enhance model performance.

## 5 A Simple, Effective SSL Baseline for VFMs

To wrap up the discussion in section 4, we propose a new semi-supervised learning method tailored to vision foundation models. Specifically, we build on self-training [64, 41, 7, 4, 49, 48], a conceptually simple SSL baseline that uses pseudo-labels from unlabeled data as additional supervision to improve the model. In the following section, we first briefly review self-training.

### 5.1 Self-training

Let us denote the predicted posterior probability of class $c$ given $\boldsymbol{x}$ as $p(c|\boldsymbol{x}; f(\boldsymbol{\theta}))$. The core idea of self-training is to gradually assign pseudo-labels to unlabeled data $\mathcal{U} = \{\boldsymbol{x}_i^u\}_{i=1}^{|\mathcal{U}|}$ based on high-confidence predictions

$$\hat{y}_i^u := \arg\max_c p\left(c \middle| \boldsymbol{x}_i^u; f_{\boldsymbol{\theta}}\right) \qquad \text{if} \ \ \max_c p(c|\boldsymbol{x}_i^u; f_{\boldsymbol{\theta}}) \geq \tau,$$

and add these pseudo-labeled data to the labeled set $\mathcal{L} = \{(\boldsymbol{x}_i^l, y_i^l)\}_{i=1}^{|\mathcal{L}|}$ for supervised learning. Here, $\boldsymbol{\theta}$ represents the weights of the current classifier, and $\tau$ is the confidence threshold. If (a) high-confidence predictions are correct and (b) the updated model gradually increases its confidence in the remaining data, self-training can be as effective as fully supervised learning on the entire dataset with true labels.

In our context, a VFM fine-tuned with PEFT using labeled dataset $\mathcal{L}$ serves as the current model to generate pseudo-labels for $\mathcal{U}$, which are then used to further fine-tune the VFM.

**Challenges.** Although self-training is conceptually simple, choosing the confidence threshold $\tau$ is difficult. A high $\tau$ excludes wrongly labeled samples but leaves too few for learning; a low $\tau$ admits

errors that reinforce bias. Moreover, the optimal $\tau$ shifts across iterations. Despite extensive study of pseudo-label selection [1, 74], most approaches remain heuristic or overly complex, undermining self-training's practicality and reliability.

**Our objective.** To leverage the high-quality pseudo-labels from label-only PEFT while circumventing the aforementioned challenges, we aim to develop a self-training-based SSL algorithm that is both simple and effective in practice. Specifically, we seek to eliminate the need for complex pseudo-label selection by utilizing all available pseudo-labels (i.e., setting $\tau = 0$). This also allows self-training to be completed in a single round, thereby removing an additional hyperparameter.

## 5.2 Ensembling of multiple PEFT and VFMs

A necessary step toward achieving our objective is to ensure pseudo-labels of even higher quality to prevent error propagation. To this end, we employ ensembling techniques [81, 19], which combine the predictions from multiple models to enhance the overall quality and reliability of pseudo-labels.

**What to Ensemble?** Unlike conventional methods that train multiple equally performant yet diverse base learners through bootstrapping, random initialization, or cyclic learning rate schedules [33], our approach exploits the unique properties of PEFT [47] and VFMs [63, 62]. Specifically, it has been observed that different PEFT methods, while achieving similar overall accuracy on downstream tasks, generate diverse predictions for individual samples. Similarly, VFMs trained on different datasets with varying objective functions exhibit diverse capabilities, with no single model demonstrating clear overall dominance.

---

**Algorithm 1:** V-PET (Figure 2)

**Input** : Labeled dataset $\mathcal{L}$ and unlabeled dataset $\mathcal{U}$
PEFT methods indexed by $n \in [1, N]$
VFMs indexed by $m \in [1, M]$
Initialized parameters of VFM on PEFT $\boldsymbol{\theta}_{n,m}$

**Output** : Optimal parameter $\boldsymbol{\theta}^*$

`// (a) Supervised Parameter Efficient`
`   Fine-Tuning`

**for** $n \in [1, N]$, $m \in [1, M]$ **do**
   Fine-tune $\boldsymbol{\theta}_{n,m}$ on $\mathcal{L}$ to obtain $\tilde{\boldsymbol{\theta}}_{n,m}$

`// (b) Pseudo-Label Generation`

**for** $n \in [1, N]$, $m \in [1, M]$ **do**
   Compute one-hot pseudo-label set $\mathcal{P}_{n,m}$ as

$$\mathcal{P}_{n,m} = \left\{ \text{one\_hot}\big(\arg\max_c f_{\tilde{\boldsymbol{\theta}}_{n,m}}(u)\big) \ \Big|\ \forall u \in \mathcal{U} \right\}.$$

`// (c) Pseudo-Label Ensemble`

Ensemble pseudo-label $\mathcal{P} := \{\bar{p}_i\}_i^{|\mathcal{U}|}$, where:

$$\bar{p}_i = \frac{1}{N \times M} \sum_{\bar{n} \in [1,N], \bar{m} \in [1,M]} \mathcal{P}_{\bar{n},\bar{m}}[i]$$

`// (d) Self-Training`

Choose $n^\star \in [1, N]$, $m^\star \in [1, M]$ and fine-tune $\boldsymbol{\theta}_{n\star,m\star}$ on $\mathcal{P}$ to get $\boldsymbol{\theta}^*$

---

**How to Ensemble?** Common strategies for ensembling different predictions are (a) *Mean Logits*—averaging class logits, and (b) *Mean Probabilities*—averaging class probabilities, both assuming similar output scales (typical with identical architectures and bootstrapped training). However, VFMs trained differently and fine-tuned via varied PEFT methods produce inconsistent scales (Figure 4), causing some models to dominate and weakening the ensemble.

To address this, we introduce a simple solution called *Mean Labels*. We first obtain predictions from each model, convert them into one-hot encoding to ensure uniformity, and average them to obtain a soft pseudo-label. Finally, we generate the augmented dataset $\mathcal{P}$ using averaged soft pseudo-label and fine-tune PEFT-based VFM models. We present our proposed pipeline in algorithm 1.

**Time Efficiency of Ensembling.** Among steps *(a)–(d)* in algorithm 1, step *(d)* dominates the wall-clock time since the pseudo-labeled set is orders of magnitude larger than the labeled one. Although our method ensembles $N \times M$ models, the overall time overhead remains marginal: step *(a)* is fast owing to the small labeled set, and steps *(b)* and *(c)* are computationally light. For instance, for V-PET, the estimated runtime is only about **1.16×** that of other SSL baselines.

# 6 Experiment

We conduct experiments to address four questions—(1) How does our method compare to existing SSL? (2) Does it scale with more PEFTs and VFMs? (3) How effective is the ensemble strategy? Following the setups in section 3 and section 4, we evaluate two variants: V-PET (ensembles over PEFT methods and VFMs) and PET (ensembles over PEFT methods on a single VFM). We use ST (self-training without ensembling) as a baseline and, in all cases, reinitialize from the original pretrained VFM before fine-tuning with pseudo-labels.

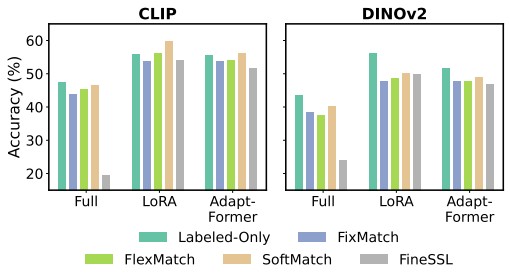

Figure 3: Average SSL accuracy with full fine-tuning or PEFT across 12 settings shows that, with fair hyper-parameter tuning, fine-tuning on limited labels can outperform SSL; PEFT boosts SSL yet matches labeled-only performance, indicating minimal unlabeled-data benefit in current VFM-based SSL (see appendix B.3).

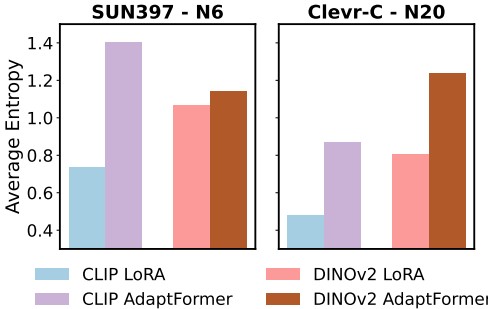

Figure 4: The average entropy of predicted probability distributions from different fine-tuned VFMs, highlighting the entropy gap among pseudo-labels, indicating poor calibration.

## 6.1 Performance Comparison

We summarize the performance comparison between ST, PET, V-PET and existing SSL methods across all the 12 settings in Table 3. Overall, V-PET achieves superior performance compared with others, as shown in the *Average* column. To further analyze the results, Figure 5 visualize the ranking frequency, where each matrix entry $(i, j)$ indicates how often method $i$ is ranked $j$-th across 12 settings. We then compute the mean rank (number in the bracket) of each method and sort them accordingly. Although V-PET does not claim first place in every single setting, it secures the top rank most frequently, showcasing the effectiveness of pseudo-label ensemble and establishing it as a simple yet powerful baseline in the VFM era.

Digging deeper into the three ST-based methods, we observe a consistent performance boost as more diverse pseudo-labels are introduced for the ensemble, i.e., ST → PET → V-PET, as shown in Figure 1. This underscores the importance of diversity among pseudo-labels.

## 6.2 Scalability of Ensembling

While most of our experiments explore ensembles of LoRA and AdaptFormer, we also examine how well our method scales when integrating additional pseudo-label sources. In particular, we expand our evaluation across two dimensions—VFMs and PEFT methods—to assess the broader applicability of our ensembling strategy.

**VFMs** Beyond ViT-B (CLIP, DINOv2), we incorporate four more VFMs, for a total of six, including four ViT-B (CLIP, DINOv2, OpenCLIP, ImangeNet21k) [15, 56] and two ViT-L (CLIP, DINOv2).

**PEFT** Following the same setup, we start with LoRA and AdaptFormer, then expand our selection with four additional PEFT: ConvPass [37], BitFit [75], VPT-Deep [36], and Fact-TT [38]. These methods span a broad range of PEFT approaches, including selective-based, adapter-based, and prompt-based techniques.

We train CLIP with LoRA on DTD N3, generate pseudo-labels, and ensemble groups of 1–6 labels. We compare (1) PEFT: ensemble of pseudo-labels from models using different PEFT methods on the same CLIP backbone, and (2) VFMs: ensemble of pseudo-labels from different VFMs all fine-tuned with LoRA. Results are averaged over five runs with distinct random pseudo-label sets (see Figure 6).

| | | DTD | | SUN397 | | RESISC45 | | Retinopathy | | CLEVR-C | | KITTI | | Average |
|---|---|---|---|---|---|---|---|---|---|---|---|---|---|---|
| | | 3 | 6 | 3 | 6 | 1 | 2 | 40 | 80 | 10 | 20 | 5 | 10 | |
| | | | | | | | | **CLIP** | | | | | | |
| **LoRA** | Labeled Only | 57.9 | 64.1 | 60.7 | 66.8 | 59.5 | 72.1 | 35.4 | 42.2 | 35.2 | 50.8 | 60.1 | 64.3 | 55.7 |
| | Fixmatch | 56.8 | 68.6 | 62.2 | **72.8** | 47.6 | 77.4 | 32.5 | 39.1 | 32.2 | 42.6 | 59.8 | 52.6 | 53.7 |
| | FlexMatch | 56.5 | 67.0 | 66.4 | 71.9 | 51.2 | 78.4 | 32.8 | 38.7 | 36.9 | **75.4** | 44.6 | 55.0 | 56.2 |
| | SoftMatch | 59.8 | 68.0 | 66.3 | 70.9 | **78.7** | **83.7** | **42.1** | 42.9 | 29.8 | 71.2 | 42.9 | 60.5 | 59.7 |
| | FineSSL | 63.1 | 69.1 | 56.3 | 61.8 | 74.2 | 83.5 | 35.0 | 30.4 | 33.6 | 38.1 | 53.7 | 48.2 | 53.9 |
| | ST | 58.9 | 63.8 | 49.2 | 52.7 | 64.1 | 77.4 | 35.7 | 42.8 | 36.5 | 51.6 | 59.8 | 64.3 | 54.7 |
| | **PET** | 63.4 | 68.2 | 65.6 | 70.8 | 67.5 | 78.6 | 36.4 | 42.4 | **38.3** | 54.6 | 60.3 | **65.7** | 59.3 |
| | **V-PET** | **65.6** | **71.7** | **67.2** | **72.8** | 66.6 | 77.2 | 38.9 | **51.1** | 36.7 | 58.5 | 58.0 | 62.3 | **60.5** |
| **AdaptFormer** | Labeled Only | 61.5 | 65.0 | 61.1 | 68.2 | 61.0 | 72.8 | 34.4 | 39.0 | 37.8 | 48.6 | 57.8 | 59.1 | 55.6 |
| | Fixmatch | 61.2 | 67.3 | 64.6 | 71.2 | 63.1 | 80.5 | 29.0 | 42.4 | 33.1 | 44.8 | 30.0 | 57.2 | 53.7 |
| | FlexMatch | 60.3 | 68.8 | 64.3 | 70.7 | 58.1 | 78.1 | 31.0 | 35.9 | 38.5 | 52.2 | 34.2 | 54.2 | 53.9 |
| | SoftMatch | 61.3 | 69.3 | 65.3 | 70.2 | 65.8 | **82.7** | 38.2 | 42.7 | 34.5 | 56.1 | 33.9 | 55.1 | 56.3 |
| | FineSSL | 61.9 | 68.1 | 58.4 | 64.9 | **73.9** | **84.2** | 25.3 | 29.8 | 26.0 | 36.9 | 42.2 | 47.1 | 51.6 |
| | ST | 62.0 | 67.2 | 58.1 | 61.3 | 68.8 | 79.3 | 34.4 | 39.6 | **39.0** | 50.3 | 58.2 | 61.9 | 56.7 |
| | **PET** | 63.8 | 68.9 | 66.8 | 71.7 | 69.5 | 80.2 | 37.1 | 41.5 | 38.4 | 55.8 | **60.5** | 61.7 | 59.7 |
| | **V-PET** | **65.7** | **71.8** | **67.8** | **73.2** | 67.9 | 78.4 | **38.9** | **51.2** | 37.0 | **59.0** | 58.2 | **63.3** | **61.0** |
| | | | | | | | | **DINOv2** | | | | | | |
| **LoRA** | Labeled Only | 61.0 | 68.4 | 59.7 | 66.8 | 48.6 | 63.1 | 40.7 | 48.0 | 35.9 | 61.7 | 46.7 | 71.7 | 56.0 |
| | Fixmatch | 57.4 | 68.4 | 56.6 | 71.4 | 36.6 | 66.7 | 38.1 | 41.2 | 35.7 | 41.0 | 14.8 | 42.9 | 47.6 |
| | FlexMatch | 52.8 | 65.4 | 55.7 | 67.7 | 33.3 | 72.6 | 36.6 | 35.8 | **37.1** | 66.9 | 26.2 | 33.6 | 48.6 |
| | SoftMatch | 53.6 | 64.0 | 60.4 | 67.3 | 46.6 | 79.5 | 32.1 | 44.5 | 33.6 | 65.1 | 27.4 | 27.9 | 50.2 |
| | FineSSL | 66.9 | 72.7 | 54.6 | 62.1 | **69.8** | **87.4** | 31.0 | 24.6 | 24.8 | 31.7 | 31.9 | 39.8 | 49.8 |
| | ST | 52.3 | 56.9 | 45.4 | 50.5 | 54.7 | 72.4 | **42.6** | 49.6 | 36.3 | 62.7 | 33.2 | **77.8** | 52.9 |
| | **PET** | 66.7 | 72.2 | 58.6 | 65.3 | 55.8 | 70.6 | 41.0 | **54.3** | 33.1 | 56.9 | 42.6 | 63.3 | 56.7 |
| | **V-PET** | **67.8** | **74.1** | **67.1** | **73.2** | 66.0 | 77.0 | 39.2 | 52.5 | 36.8 | 59.3 | **58.2** | 63.7 | **61.2** |
| **AdaptFormer** | Labeled Only | 63.0 | 67.6 | 58.9 | 67.0 | 47.0 | 57.4 | 34.4 | 53.1 | 27.3 | 41.8 | 52.3 | 49.8 | 51.6 |
| | Fixmatch | 59.5 | 70.4 | 57.9 | 69.7 | 26.4 | 62.3 | 38.7 | 45.6 | 32.7 | 38.9 | 19.8 | 50.4 | 47.7 |
| | FlexMatch | 57.1 | 65.9 | 56.0 | 67.5 | 51.3 | 64.0 | 33.4 | 40.2 | 33.4 | 42.2 | 34.3 | 27.9 | 47.8 |
| | SoftMatch | 58.7 | 66.2 | 60.0 | 66.7 | 53.1 | **78.1** | 33.5 | 32.2 | 32.9 | 45.0 | 24.5 | 30.9 | 48.5 |
| | FineSSL | 64.0 | 71.2 | 56.0 | 63.3 | 62.3 | 69.7 | 17.7 | 24.3 | 25.8 | 30.0 | 31.8 | 45.0 | 46.8 |
| | ST | 62.3 | 67.0 | 59.0 | 67.1 | 54.1 | 66.2 | 34.9 | **56.0** | 27.4 | 43.1 | 43.9 | 48.7 | 52.5 |
| | **PET** | 67.2 | 73.0 | 63.5 | 70.9 | 56.2 | 72.0 | **40.2** | 55.5 | 33.1 | 58.0 | 45.2 | 60.9 | 58.0 |
| | **V-PET** | **68.2** | **74.2** | **67.7** | **73.4** | **66.3** | 78.0 | 39.4 | 52.5 | **36.8** | **60.0** | **59.2** | **61.7** | **61.5** |

Table 3: Performance (%) comparison of baselines, existing SSL approaches, and our proposed methods on six diverse datasets (12 settings). The best result within each PEFT and VFM is highlighted in **bold**. We report the full results in appendix B.2.

**Mean Label Consistently Outperforms.** As also shown in Figure 6, we investigate three ensemble strategies: (1) **Mean Label** (our proposed); (2) **Mean Logits**; and (3) **Mean Probabilities**. We find that the Mean Label strategy is the most effective among the three. It consistently outperforms the other two strategies across different ensemble sources and ensemble sizes. This suggests that the Mean Label approach better captures the diversity of pseudo-labels and maximizes the information utilization from the ensemble sources.

**Both PEFT and VFMs Help.** Whether we ensemble the pseudo-labels generated across different PEFT or VFMs, the performance is consistently improved, indicating that both PEFT and VFMs contribute to the diversity of pseudo-labels and improve the ensemble robustness.

**Diminishing Return.** The performance improvement diminishes as we increase the number of ensemble sources, suggesting the performance gain from ensemble is not linearly scalable. In order to achieve a balance between performance and computational cost, we recommend using around 2 to 4 in this case.

**Impact of Pseudo-Label Quality.** When applying V-PET, one might wonder: if the performances of different VFMs vary significantly, can their pseudo-labels still be ensembled to boost downstream results? The answer is yes, we demonstrate the result in appendix B.6.

## 6.3 The Effectiveness of Hyperparameter Tuning

We also investigated whether the proposed hyperparameter tuning strategy accurately gauges SSL methods' performance. To illustrate its advantages, we compare it with two baselines, both measured via the absolute difference from the oracle test performance. Specifically, the oracle test performance is defined as the highest accuracy across the entire hyperparameter search space; each method selects a

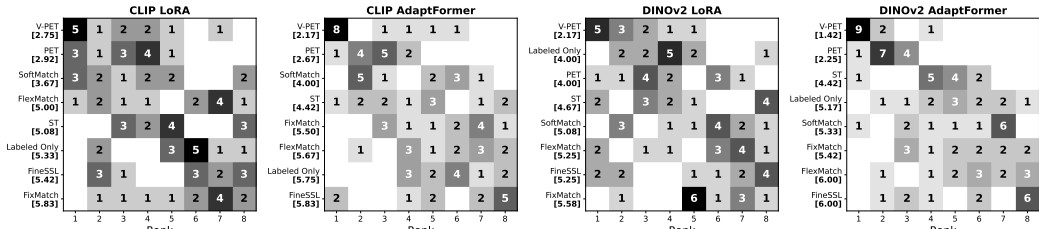

Figure 5: Ranking frequency across SSL methods by the proposed benchmark. The number in $(i, j)$ indicates the frequency of method $i$ is ranked $j$-th across 12 settings. The number in brackets indicates the average rank, where the higher rank is better.

Table 4: Hyperparameter tuning comparison: our method versus baselines (random or chosen by one criterion). The absolute errors (%) from oracle test accuracy (smaller the better) are reported averaged across 12 datasets, 5 SSL baselines, and 2 VFMs trained on 2 PEFT. The reported values are in the format: mean $\pm$ standard deviation.

| RankMe | AMI | ARI | V-Measure | FMI | CHI | BNM | Random | Ours |
|--------|-----|-----|-----------|-----|-----|-----|--------|------|
| 7.9±8.6 | 2.8±5.0 | 2.8±4.9 | 2.8±4.9 | 2.9±4.9 | 3.7±7.0 | 12.0±12.1 | 8.9±8.1 | **2.6±4.3** |

hyperparameter configuration, and we record how far its resulting accuracy is from this optimum. By taking the absolute value of this difference, we quantify how closely each tuning method approximates the best possible result.

Our baselines are: (1) *Random expectation*, which represents the average performance obtained by randomly choosing hyperparameters from the search space, and (2) *Single criterion*, which selects the best hyperparameter configuration using only one metric. Comparing with (1) offers insight into how well our method leverages the search space relative to a purely random choice, while comparing with (2) illustrates the advantage of integrating multiple evaluation criteria. As shown in Table 4, we measure the average absolute error between each method's chosen accuracy and the oracle's best accuracy (i.e., the maximum accuracy achievable in the search space) across 12 datasets, 5 SSL baselines, 2 VFMs (CLIP and DINOv2), and 2 PEFT methods (LoRA and AdaptFormer). Our method outperforms both baselines in most cases, suggesting that combining multiple metrics produces a more reliable and robust assessment of SSL methods. We hope future research adopts this approach to strengthen SSL evaluations.

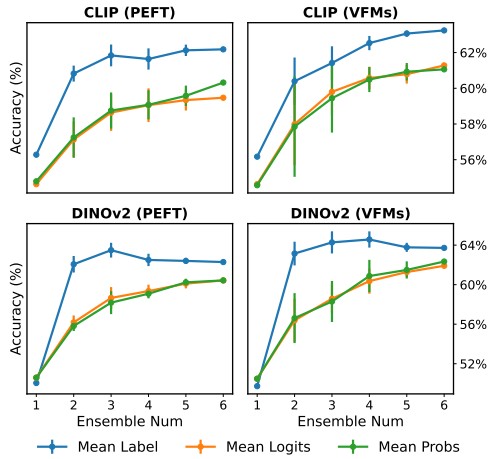

Figure 6: Scaling analysis for ensembling more PEFT methods and VFMs. Starting from LoRA fine-tuning, performance improves with diverse pseudo-labels, but the gain diminishes with more ensemble sources. *Mean Label* consistently outperforms *Mean Logits* and *Mean Probabilities*.

## 7 Conclusion

We conduct a comprehensive and systematic empirical study for SSL in the era of VFMs, revealing that VFMs can still benefit from SSL but require a specifically tailored approach. In particular, we propose a simple yet highly effective self-training algorithm that ensembles pseudo-labels from multiple VFMs and parameter-efficient fine-tuning (PEFT) techniques. Overall, our study offers actionable insights into SSL with VFMs and lays the groundwork for more scalable and practical semi-supervised learning in the era of foundation models.

## Acknowledgment

This research is supported by grants from the National Science Foundation (ICICLE: OAC-2112606). We appreciate the generous support of computational resources from the Ohio Supercomputer Center (OSC).

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

The appendix is organized as follows:

- appendix A provides details about the datasets used in our experiments.

- appendix B presents details about our experiments and additional experiments we conducted.

- appendix C provides a detailed description of the hyperparameter search algorithms we proposed and metrics considered for the experiments in the main paper.

## A    Dataset Details

We select 6 datasets with a diverse ranges of applications for our experiments, including:

- **DTD** [16]: The Describable Texture Dataset (DTD) is designed for texture pattern recognition. It includes a diverse collection of images from 47 distinct texture categories, offering rich and varied examples to evaluate texture feature extraction and classification methods under natural conditions.

- **SUN397** [69]: SUN397 is a large-scale scene recognition benchmark that covers 397 categories of both indoor and outdoor scenes. It provides a comprehensive testbed for scene classification algorithms.

- **Resisc45** [14]: Resisc45 is a remote sensing image dataset specifically created for scene classification tasks. It comprises thousands of images across 45 categories, representing diverse landscapes such as urban, rural, and industrial areas. This dataset is instrumental in developing and benchmarking algorithms for geographic information systems and environmental monitoring.

- **Diabetic-Retinopathy** [22]: This dataset focuses on high-resolution retinal images used for detecting diabetic retinopathy. It includes images that span various stages of the disease, providing a robust resource for training and evaluating deep learning models aimed at early diagnosis and automated medical analysis in ophthalmology.

- **Clevr-Count** [39]: Derived from the CLEVR family of datasets, Clevr-Count is centered on visual reasoning, particularly object counting tasks. Using synthetically generated images, it challenges models to accurately count objects within complex scenes, thereby assessing their ability to understand spatial relationships and compositional structures in visual data.

- **KITTI-Dist** [27]: KITTI-Dist is based on the renowned KITTI benchmark and is tailored for depth estimation in vehicular contexts. It provides high-quality stereo images along with ground-truth depth maps, supporting research in autonomous driving, 3D reconstruction, and stereoscopic vision, and serves as a key resource for evaluating depth estimation algorithms.

The detailed statistics of the datasets, specifically the validation and test set sizes, are presented in Table 5.

| Dataset | $N^v$ | Test Size |
|---|---|---|
| DTD | 752 | 1,880 |
| SUN397 | 17,401 | 21,750 |
| Resisc45 | 5,040 | 6,300 |
| Retinopathy | 9,207 | 42670 |
| Clevr-C | 14,000 | 15,000 |
| KITTI | 1,354 | 711 |

Table 5: Dataset statistics.

## B    Experiment Details

In this section, we provide additional details on the hyperparameters and computation settings used in our experiments.

### B.1 Computation Details

Our experiments were conducted on a workstation equipped with eight NVIDIA RTX 6000 Ada GPUs, two AMD EPYC 9554 64-Core Processors, and 800GB of RAM. Additionally, we utilized NVIDIA Tesla V100, NVIDIA Tesla A100, and NVIDIA RTX H100 GPUs for certain experiments. All experiments were implemented using PyTorch [54].

### B.2 SSL and Labeled-Only Baseline Details

For all the experiments, we employ AdamW [44] optimizer with a cosine annealing learning rate scheduler [43] with warm-up period with $2.5\%$ of total iterations. We use a batch size of $32 + 32$ for all experiments, where the first 32 corresponds to labeled data and the second 32 corresponds to unlabeled data. We list the hyperparameter search spaces for the SSL, Labeled-Only, and ST settings in Table 6, including the drop path rate (dpr) [34], training augmentation (train-aug), LoRA dimension (lora_dim), adapter bottleneck size (adapter_bottleneck), weight decay, momentum, learning rate (lr), and number of epochs. For other unmentioned SSL algorithm related hyperparameters, we use the default values provided in the original papers.

| Hyperparameter | SSL | Labeled-Only | ST |
|---|---|---|---|
| dpr | 0 | [0, 0.2] | 0 |
| train-aug | - | [weak, strong] | weak |
| lora_dim | 4 | [4, 16] | 4 |
| adapter_bottleneck | 16 | [4, 16] | 16 |
| weight_decay | 5e-4 | 5e-4 | 5e-4 |
| momentum | 0.9 | 0.9 | 0.9 |
| lr | Full FT: [1e-4, 1e-5, 1e-6] PEFT: [1e-3, 1e-4, 1e-5] | Full FT: [1e-4, 1e-5, 1e-6] PEFT: [1e-3, 1e-4, 1e-5] | Full FT: [1e-4, 1e-5, 1e-6] PEFT: [1e-3, 1e-4, 1e-5] |
| epochs | Full FT: 35 PEFT: 30 | Full FT: 60 PEFT: 50 | Full FT: 35 PEFT: 30 |

Table 6: Hyperparameter search spaces for SSL, Labeled-Only, and ST settings.

As a completion of Table 3, we present the detailed results of the SSL baseline experiment with different SSL methods, including full fine-tuning, in Table 7.

### B.3 PEFT Experiment Details

As a completion of Figure 3, we present the detailed results of the PEFT experiment with different SSL methods in Table 8. We present the detailed results of the PEFT experiment with different SSL methods in Table 8.

### B.4 Linear Probing with Different Classification Heads

We provide results of frozen backbone with different classification heads on our benchmark in Table 9.

### B.5 Impact of Labeled-Only Initialization on SSL

Per discussion in section 4, a natural question is whether the performance of SSL methods can be improved by initializing from a model trained on labeled data using PEFT. To explore this, we conduct experiments on three datasets: DTD (N6), Diabetic Retinopathy (N80), and KITTI (N10). We first train DINOv2 models on labeled data using LoRA and then fine-tune them with SoftMatch. These datasets were chosen because their labeled-only performance exceeds that of SoftMatch by a large margin, as shown in Table 3.

The results, summarized in Figure 10, suggest that even when starting from the labeled-only trained model, SSL performance still falls short. In some cases, it performs even worse than training from scratch.

| | | DTD | | Sun397 | | Resisc45 | | Retinopathy | | Clevr-C | | KITTI | | AVERAGE |
|---|---|---|---|---|---|---|---|---|---|---|---|---|---|---|
| | | 3 | 6 | 3 | 6 | 1 | 2 | 40 | 80 | 10 | 20 | 5 | 10 | |
| **CLIP** | | | | | | | | | | | | | | |
| **Full** | Labeled-Only | 53.46% | 62.82% | 52.79% | 59.49% | 52.98% | 65.51% | 31.78% | 32.69% | 25.03% | 29.62% | 37.55% | 65.54% | 47.44% |
| | Fixmatch | 46.97% | 60.37% | 45.49% | 60.97% | 31.76% | 74.48% | 24.64% | 45.95% | 21.70% | 21.05% | 45.43% | 45.15% | 43.66% |
| | FlexMatch | 50.69% | 61.06% | 44.41% | 61.35% | 41.62% | 79.05% | 28.35% | 38.12% | 25.31% | 20.75% | 44.87% | 47.40% | 45.25% |
| | SoftMatch | 52.29% | 62.55% | 50.69% | 60.70% | 59.86% | 84.67% | 32.91% | 24.15% | 19.53% | 20.02% | 31.22% | 57.52% | 46.34% |
| | FineSSL | 7.02% | 17.18% | 1.46% | 4.01% | 7.78% | 8.37% | 39.64% | 16.97% | 19.53% | 26.08% | 41.49% | 44.59% | 19.51% |
| | ST | 54.10% | 61.60% | 52.88% | 59.30% | 55.95% | 69.90% | 34.69% | 32.95% | 24.65% | 29.78% | 34.46% | 65.40% | 47.97% |
| | PET | 63.19% | 68.67% | 63.28% | 68.83% | 68.32% | 79.73% | 39.46% | 43.35% | 37.33% | 49.87% | 60.34% | 63.85% | 58.85% |
| | V-PET | 66.33% | 70.64% | 66.42% | 72.23% | 67.49% | 79.11% | 39.46% | 55.01% | 35.61% | 54.88% | 57.24% | 65.12% | 60.80% |
| **LoRA** | Labeled-Only | 57.87% | 64.10% | 60.66% | 66.79% | 59.48% | 72.05% | 35.38% | 42.16% | 35.21% | 50.80% | 60.06% | 64.28% | 55.74% |
| | Fixmatch | 56.81% | 68.56% | 62.17% | 72.78% | 47.56% | 77.41% | 32.45% | 39.09% | 32.20% | 42.61% | 59.77% | 52.60% | 53.67% |
| | FlexMatch | 56.49% | 66.97% | 66.39% | 71.90% | 51.24% | 78.40% | 32.84% | 38.67% | 36.89% | 75.35% | 44.59% | 54.99% | 56.23% |
| | SoftMatch | 59.79% | 68.03% | 66.31% | 70.87% | 78.70% | 83.65% | 42.05% | 42.93% | 29.78% | 71.20% | 42.90% | 60.48% | 59.72% |
| | FineSSL | 63.14% | 69.10% | 56.25% | 61.78% | 74.17% | 83.48% | 35.03% | 30.42% | 33.55% | 38.07% | 53.73% | 48.24% | 53.91% |
| | ST | 58.94% | 63.83% | 49.24% | 52.68% | 64.05% | 77.43% | 35.73% | 42.77% | 36.45% | 51.63% | 59.77% | 64.28% | 54.73% |
| | PET | 63.35% | 68.24% | 65.61% | 70.80% | 67.52% | 78.56% | 36.42% | 42.39% | 38.27% | 54.63% | 60.34% | 65.68% | 59.32% |
| | V-PET | 65.59% | 71.65% | 67.19% | 72.83% | 66.56% | 77.21% | 38.85% | 51.06% | 36.69% | 58.49% | 57.95% | 62.31% | 60.53% |
| **AdaptFormer** | Labeled-Only | 61.54% | 64.95% | 61.14% | 68.23% | 60.98% | 72.83% | 34.38% | 38.99% | 37.80% | 48.63% | 57.81% | 59.07% | 55.53% |
| | Fixmatch | 61.22% | 67.34% | 64.63% | 71.19% | 63.06% | 80.49% | 28.96% | 42.41% | 33.13% | 44.76% | 29.96% | 57.24% | 53.70% |
| | FlexMatch | 60.27% | 68.78% | 64.32% | 70.73% | 58.13% | 78.08% | 31.02% | 35.93% | 38.50% | 52.21% | 34.18% | 54.15% | 53.86% |
| | SoftMatch | 61.33% | 69.31% | 65.33% | 70.17% | 65.78% | 82.67% | 38.16% | 42.66% | 34.48% | 56.08% | 33.90% | 55.13% | 56.25% |
| | FineSSL | 61.91% | 68.14% | 58.37% | 64.86% | 73.95% | 84.17% | 25.27% | 29.78% | 25.97% | 36.87% | 42.19% | 47.12% | 51.55% |
| | ST | 61.97% | 67.23% | 58.10% | 61.29% | 68.84% | 79.33% | 34.42% | 39.60% | 39.01% | 50.33% | 58.23% | 61.88% | 56.69% |
| | PET | 63.78% | 68.94% | 66.78% | 71.71% | 69.54% | 80.16% | 37.14% | 41.51% | 38.45% | 55.78% | 60.48% | 61.74% | 59.67% |
| | V-PET | 65.69% | 71.76% | 67.81% | 73.23% | 67.92% | 78.43% | 38.90% | 51.22% | 36.96% | 59.01% | 58.23% | 63.29% | 61.04% |
| **DINOv2** | | | | | | | | | | | | | | |
| **Full** | Labeled-Only | 46.44% | 61.76% | 47.58% | 58.83% | 5.22% | 56.95% | 34.24% | 41.54% | 26.47% | 26.94% | 53.87% | 62.31% | 43.51% |
| | Fixmatch | 49.57% | 63.19% | 42.98% | 64.73% | 20.97% | 55.16% | 26.96% | 20.55% | 16.75% | 16.82% | 31.08% | 50.49% | 38.27% |
| | FlexMatch | 44.04% | 60.80% | 44.97% | 58.23% | 24.71% | 63.75% | 23.78% | 26.09% | 17.14% | 15.80% | 40.37% | 30.24% | 37.49% |
| | SoftMatch | 27.02% | 57.55% | 48.73% | 59.06% | 51.52% | 77.41% | 26.28% | 36.17% | 16.54% | 17.71% | 16.17% | 46.55% | 40.06% |
| | FineSSL | 23.78% | 48.40% | 2.76% | 16.57% | 10.54% | 22.05% | 22.25% | 21.17% | 20.75% | 23.52% | 39.94% | 35.44% | 23.93% |
| | ST | 49.47% | 60.43% | 46.83% | 56.59% | 5.38% | 61.79% | 35.68% | 43.38% | 27.17% | 26.81% | 60.20% | 60.90% | 44.55% |
| | PET | 64.41% | 69.84% | 59.94% | 66.87% | 55.76% | 69.62% | 34.34% | 56.55% | 31.40% | 55.02% | 46.98% | 64.98% | 56.31% |
| | V-PET | 66.97% | 73.78% | 65.53% | 71.48% | 67.32% | 79.25% | 36.55% | 52.98% | 35.93% | 56.55% | 55.98% | 64.70% | 60.59% |
| **LoRA** | Labeled-Only | 61.01% | 68.40% | 59.71% | 66.78% | 48.59% | 63.08% | 40.69% | 48.02% | 35.87% | 61.72% | 46.69% | 71.73% | 56.02% |
| | Fixmatch | 57.39% | 68.40% | 56.62% | 71.38% | 36.60% | 66.65% | 38.11% | 41.16% | 35.65% | 40.99% | 14.77% | 42.90% | 47.55% |
| | FlexMatch | 52.82% | 65.37% | 55.69% | 67.65% | 33.32% | 72.63% | 36.56% | 35.81% | 37.13% | 66.85% | 26.16% | 33.61% | 48.63% |
| | SoftMatch | 53.62% | 64.04% | 60.41% | 67.34% | 46.60% | 79.54% | 32.05% | 44.48% | 33.57% | 65.07% | 27.43% | 27.85% | 50.17% |
| | FineSSL | 66.97% | 72.66% | 54.62% | 62.07% | 69.78% | 87.38% | 30.99% | 24.55% | 24.77% | 31.73% | 31.93% | 39.80% | 49.77% |
| | ST | 52.34% | 56.86% | 45.39% | 50.47% | 54.73% | 72.44% | 42.58% | 49.56% | 36.29% | 62.74% | 33.19% | 77.78% | 52.86% |
| | PET | 66.65% | 72.18% | 58.57% | 65.29% | 55.79% | 70.59% | 40.99% | 54.28% | 33.11% | 56.85% | 42.62% | 63.29% | 56.68% |
| | V-PET | 67.77% | 74.10% | 67.12% | 73.24% | 66.03% | 77.03% | 39.20% | 52.48% | 36.77% | 59.25% | 58.23% | 63.71% | 61.24% |
| **AdaptFormer** | Labeled-Only | 62.98% | 67.61% | 58.87% | 67.03% | 47.00% | 57.41% | 34.42% | 53.07% | 27.33% | 41.79% | 52.32% | 49.79% | 51.64% |
| | Fixmatch | 59.47% | 70.37% | 57.85% | 69.68% | 26.41% | 62.30% | 38.73% | 45.55% | 32.65% | 38.89% | 19.83% | 50.35% | 47.67% |
| | FlexMatch | 57.07% | 65.90% | 56.01% | 67.53% | 51.29% | 64.02% | 33.43% | 40.20% | 33.35% | 42.16% | 34.32% | 27.85% | 47.76% |
| | SoftMatch | 58.67% | 66.17% | 59.97% | 66.65% | 53.06% | 78.06% | 33.47% | 32.17% | 32.91% | 45.00% | 24.47% | 30.94% | 48.46% |
| | FineSSL | 64.04% | 71.17% | 55.96% | 63.34% | 62.32% | 69.73% | 17.66% | 24.30% | 25.79% | 30.03% | 31.79% | 45.01% | 46.76% |
| | ST | 62.29% | 67.02% | 59.00% | 67.11% | 54.06% | 66.17% | 34.87% | 55.96% | 27.41% | 43.13% | 43.88% | 48.66% | 52.46% |
| | PET | 67.18% | 72.98% | 63.47% | 70.93% | 56.21% | 72.00% | 40.18% | 55.53% | 33.05% | 57.95% | 45.15% | 60.90% | 57.96% |
| | V-PET | 68.24% | 74.15% | 67.65% | 73.43% | 66.32% | 77.98% | 39.39% | 52.53% | 36.77% | 59.97% | 59.21% | 61.74% | 61.45% |

Table 7: Complete results of the SSL baseline experiment with different SSL methods.

Table 10: Performance of starting from a labeled-only model then fine-tuning with SoftMatch. Results on DTD (N6), Retinopathy (N80), and KITTI (N10). LoRA dim=4 is fixed.

| Dataset | Labeled-Only | SoftMatch | Labeled-Only → SoftMatch |
|---|---|---|---|
| DTD N6 | **68.4%** | 64.0% | 63.5% |
| Retinopathy N80 | **44.9%** | 44.5% | 43.9% |
| KITTI N10 | **54.0%** | 27.9% | 50.1% |

## B.6 The Effectiveness of VFMs Ensemble

To assess the benefit of pseudo-label ensembling across VFMs, even when their individual accuracies differ significantly, we performed an ablation study on the CLIP-B model. We trained three variants on different pseudo-label sets: (1) labels generated by CLIP itself, (2) labels from ViT-B-IN21k, a comparatively weaker VFM, and (3) the ensemble of both sources. Results in Table 11 show that, despite ViT-B-IN21k's lower standalone accuracy, combining its labels with CLIP's improves overall performance.

## C Hyperparameter Tuning Details

We provide the detailed hyperparameter tuning procedure in this section.

| | | | DTD | | Sun397 | | Resisc45 | | Retinopathy | | Clevr-C | | KITTI | |
|---|---|---|---|---|---|---|---|---|---|---|---|---|---|---|
| | | | N3 | N6 | N3 | N6 | N1 | N2 | N40 | N80 | N10 | N20 | N5 | N10 |
| **CLIP** | Labeled-Only | Full | 53.46% | 62.82% | 52.79% | 59.49% | 52.98% | 65.51% | 31.78% | 32.69% | 25.03% | 29.62% | 37.55% | 65.54% |
| | | LoRA | 57.87% | 64.10% | 60.66% | 66.79% | 59.48% | 72.05% | 35.38% | 42.16% | 35.21% | 50.80% | 60.06% | 64.28% |
| | | AdaptFormer | 61.54% | 64.95% | 61.14% | 68.23% | 60.98% | 72.83% | 34.38% | 38.99% | 37.80% | 48.63% | 57.81% | 59.07% |
| | | Δ | 6.25% | 1.70% | 8.11% | 8.02% | 7.25% | 6.93% | 3.10% | 7.88% | 11.47% | 20.10% | 21.38% | -3.87% |
| | FixMatch | Full | 46.97% | 60.37% | 45.49% | 60.97% | 31.76% | 74.48% | 24.64% | 45.95% | 21.70% | 21.05% | 45.43% | 45.15% |
| | | LoRA | 56.81% | 68.56% | 62.17% | 72.78% | 47.56% | 77.41% | 32.45% | 39.09% | 32.20% | 42.61% | 59.77% | 52.60% |
| | | AdaptFormer | 61.22% | 67.34% | 64.63% | 71.19% | 63.06% | 80.49% | 28.96% | 42.41% | 33.13% | 44.76% | 29.96% | 57.24% |
| | | Δ | 12.05% | 7.58% | 17.92% | 11.01% | 23.55% | 4.48% | 6.07% | -5.20% | 10.97% | 22.63% | -0.56% | 9.77% |
| | FlexMatch | Full | 50.69% | 61.06% | 44.41% | 61.35% | 41.62% | 79.05% | 28.35% | 38.12% | 25.31% | 20.75% | 44.87% | 47.40% |
| | | LoRA | 56.49% | 66.97% | 66.39% | 71.90% | 51.24% | 78.40% | 32.84% | 38.67% | 36.89% | 75.35% | 44.59% | 54.99% |
| | | AdaptFormer | 60.27% | 68.78% | 64.32% | 70.73% | 58.13% | 78.08% | 31.02% | 35.93% | 38.50% | 52.21% | 34.18% | 54.15% |
| | | Δ | 7.69% | 6.81% | 20.94% | 9.97% | 13.06% | -0.81% | 3.58% | -0.82% | 12.39% | 43.03% | -5.49% | 7.17% |
| | SoftMatch | Full | 52.29% | 62.55% | 50.69% | 60.70% | 59.86% | 84.67% | 32.91% | 24.15% | 19.53% | 20.02% | 31.22% | 57.52% |
| | | LoRA | 59.79% | 68.03% | 66.31% | 70.87% | 78.70% | 83.65% | 42.05% | 42.93% | 29.78% | 71.20% | 42.90% | 60.48% |
| | | AdaptFormer | 61.33% | 69.31% | 65.33% | 70.17% | 65.78% | 82.67% | 38.16% | 42.66% | 34.48% | 56.08% | 33.90% | 55.13% |
| | | Δ | 8.27% | 6.12% | 15.13% | 9.82% | 12.38% | -1.51% | 7.19% | 18.65% | 12.60% | 43.62% | 7.17% | 0.28% |
| | FineSSL | Full | 7.02% | 17.18% | 1.46% | 4.01% | 7.78% | 8.37% | 39.64% | 16.97% | 19.53% | 26.08% | 41.49% | 44.59% |
| | | LoRA | 63.14% | 69.10% | 56.25% | 61.78% | 74.17% | 83.48% | 35.03% | 30.42% | 33.55% | 38.07% | 53.73% | 48.24% |
| | | AdaptFormer | 61.91% | 68.14% | 58.37% | 64.86% | 73.95% | 84.17% | 25.27% | 29.78% | 25.97% | 36.87% | 42.19% | 47.12% |
| | | Δ | 55.51% | 51.44% | 55.85% | 59.31% | 66.29% | 75.46% | -9.49% | 13.13% | 10.23% | 11.39% | 6.47% | 3.09% |
| **DINOv2** | Labeled-Only | Full | 46.44% | 61.76% | 47.58% | 58.83% | 5.22% | 56.95% | 34.24% | 41.54% | 26.47% | 26.94% | 53.87% | 62.31% |
| | | LoRA | 61.01% | 68.40% | 59.71% | 66.78% | 48.59% | 63.08% | 40.69% | 48.02% | 35.87% | 61.72% | 46.69% | 71.73% |
| | | AdaptFormer | 62.98% | 67.61% | 58.87% | 67.03% | 47.00% | 57.41% | 34.42% | 53.07% | 27.33% | 41.79% | 52.32% | 49.79% |
| | | Δ | 15.56% | 6.25% | 11.71% | 8.07% | 42.57% | 3.29% | 3.32% | 9.00% | 5.13% | 24.82% | -4.36% | -1.55% |
| | FixMatch | Full | 49.57% | 63.19% | 42.98% | 64.73% | 20.97% | 55.16% | 26.96% | 20.55% | 16.75% | 16.82% | 31.08% | 50.49% |
| | | LoRA | 57.39% | 68.40% | 56.62% | 71.38% | 36.60% | 66.65% | 38.11% | 41.16% | 35.65% | 40.99% | 14.77% | 42.90% |
| | | AdaptFormer | 59.47% | 70.37% | 57.85% | 69.68% | 26.41% | 62.30% | 38.73% | 45.55% | 32.65% | 38.89% | 19.83% | 50.35% |
| | | Δ | 8.86% | 6.20% | 14.25% | 5.80% | 10.54% | 9.32% | 11.45% | 22.81% | 17.40% | 23.12% | -13.78% | -3.87% |
| | FlexMatch | Full | 44.04% | 60.80% | 44.97% | 58.23% | 24.71% | 63.75% | 23.78% | 26.09% | 17.14% | 15.80% | 40.37% | 30.24% |
| | | LoRA | 52.82% | 65.37% | 55.69% | 67.65% | 33.32% | 72.63% | 36.56% | 35.81% | 37.13% | 66.85% | 26.16% | 33.61% |
| | | AdaptFormer | 57.07% | 65.90% | 56.01% | 67.53% | 51.29% | 64.02% | 33.43% | 40.20% | 33.35% | 42.16% | 34.32% | 27.85% |
| | | Δ | 10.90% | 4.84% | 10.89% | 9.36% | 17.59% | 4.58% | 11.21% | 11.92% | 18.10% | 38.71% | -10.13% | 0.49% |
| | SoftMatch | Full | 27.02% | 57.55% | 48.73% | 59.06% | 51.52% | 77.41% | 26.28% | 36.17% | 16.54% | 17.71% | 16.17% | 46.55% |
| | | LoRA | 53.62% | 64.04% | 60.41% | 67.34% | 46.60% | 79.54% | | | 33.57% | 65.07% | 27.43% | 27.85% |
| | | AdaptFormer | 58.67% | 66.17% | 59.97% | 66.65% | 53.06% | 78.06% | 33.47% | 32.17% | 32.91% | 45.00% | 24.47% | 30.94% |
| | | Δ | 29.12% | 7.55% | 11.47% | 7.94% | -1.69% | 1.39% | 6.48% | 2.16% | 16.70% | 37.32% | 9.77% | -17.16% |
| | FineSSL | Full | 23.78% | 48.40% | 2.76% | 16.57% | 10.54% | 22.05% | 22.25% | 21.17% | 20.75% | 23.52% | 39.94% | 35.44% |
| | | LoRA | 66.97% | 72.66% | 54.62% | 62.07% | 69.78% | 87.38% | 30.99% | 24.55% | 24.77% | 31.73% | 31.93% | 39.80% |
| | | AdaptFormer | 64.04% | 71.17% | 55.96% | 63.34% | 62.32% | 69.73% | 17.66% | 24.30% | 25.79% | 30.03% | 31.79% | 45.01% |
| | | Δ | 41.73% | 23.51% | 52.53% | 46.13% | 55.51% | 56.51% | 2.08% | 3.25% | 4.53% | 7.36% | -8.09% | 6.96% |

Table 8: Performance comparison of various SSL methods on our proposed benchmark datasets. For each dataset, model, and SSL baseline, we report the arithmetic mean of test accuracy across different PEFT, subtracted by the test accuracy of full fine-tuning. A positive value indicates that PEFT outperform full fine-tuning in the given scenario. Overall, the results show that PEFT consistently outperform full fine-tuning, with very few exceptions. This demonstrates the effectiveness of PEFT in enhancing the performance of VFMs within SSL frameworks.

| N | DTD 6 | SUN397 6 | Resisc45 2 | Retinopathy 80 | Clevr-C 20 | KITTI 10 |
|---|---|---|---|---|---|---|
| CLIP-LINEAR | 61.80% | 63.70% | 69.30% | 35.90% | 33.10% | 51.10% |
| CLIP-MLP-96 | 63.09% | 60.29% | 67.89% | 36.91% | 38.11% | 50.77% |
| CLIP-MLP-768 | 62.29% | 62.52% | 69.62% | 39.67% | 39.51% | 48.10% |
| CLIP LoRA | 64.10% | 66.79% | 72.05% | 42.16% | 50.80% | 64.28% |
| DINOv2-LINEAR | 66.70% | 65.30% | 52.40% | 41.20% | 30.50% | 51.30% |
| DINOv2-MLP-96 | 66.97% | 62.41% | 50.30% | 44.33% | 36.37% | 50.91% |
| DINOv2-MLP-768 | 67.98% | 61.16% | 52.06% | 35.66% | 35.21% | 50.77% |
| DINOv2 LoRA | 68.40% | 66.78% | 63.08% | 48.02% | 61.72% | 71.73% |

Table 9: Linear probing results compared to LoRA under various scenarios, where LINEAR denote a single linear layer and MLP-x denotes an MLP with a single hidden layer of size x. The results demonstrate that LoRA is a more effective approach for fine-tuning vision foundation models.

## C.1 Hyperparameter Tuning Procedure

Concretely, given a set of hyperparameters $\{\phi_1, \phi_2, \ldots, \phi_h\}$, and models trained with them $\{f_{(\theta,\phi_1)}, f_{(\theta,\phi_2)}, \ldots, f_{(\theta,\phi_h)}\}$, a held-out unlabled validation set $\mathcal{D}^v = \{(x_i^v)\}_{i=1}^{N^v}$ is use to calculate the 7 unsupervised criteria. Then we calculate the rank of each criterion for each hyperparameter set. We will choose the hyperparameter set with the lowest average rank $\frac{1}{7}\{R\}$ across all criteria.

Table 11: We use pseudo-labels from different source to train a CLIP-B model. Demonstrating when performing ensembling training, even weaker pseudo-labels will boost performance.

| Model | Test Accuracy |
|---|---|
| CLIP | 56.17% |
| ViT-B-IN21k | 25.27% |
| CLIP + ViT-B-IN21k | **58.62%** |

**Problem Formulation** Formally, the objective is to find a suitable function,

$$
\begin{aligned}
f &:= g \circ h \in \mathbb{R}^i \to \mathbb{R}^j && \text{(model)}, \\
h &\in \mathbb{R}^i \to \mathbb{R}^k && \text{(feature extractor)}, \\
g &\in \mathbb{R}^k \to \mathbb{R}^j && \text{(linear projector)}.
\end{aligned}
$$

where $h$ serves as a feature extractor, and $g$ is a linear projector. The aim is to use $\mathcal{L}$ and $\mathcal{U}$ to learn $h$ and $g$ that minimize the generalization error on the test set $\mathcal{T}$.

For any dataset $\mathcal{D} := x_i{}_{i=1}^{|\mathcal{D}|}$, regardless of whether it is labeled, we denote the output of $h$ as **features**, the output of $g$ as **logits**, and the argmax of the logits as **predictions**.

$$
\begin{aligned}
\phi_{\mathcal{D}} &:= \{h(x_i)\}_{i=1}^{|\mathcal{D}|} && \text{(features)}, \\
\ell_{\mathcal{D}} &:= \{g(h(x_i))\}_{i=1}^{|\mathcal{D}|} && \text{(logits)}, \\
\pi_{\mathcal{D}} &:= \{\arg\max g(h(x_i))\}_{i=1}^{|\mathcal{D}|} && \text{(predictions)}.
\end{aligned}
$$

**Tuning without Labels** The primary challenge in tuning hyperparameters within an unsupervised setting lies in effectively adjusting them without access to labels. Inspired by the approach for standardizing hyperparameter tuning in Unsupervised Domain Adaptation (UDA) [23], we propose tuning the hyperparameters using $\phi_{\mathcal{D}_v}$, $\ell_{\mathcal{D}_v}$, and $\pi_{\mathcal{D}_v}$. Specifically, we define function $v$ as a **validator**:

$$
v := (\phi_{\mathcal{D}_v}, \ell_{\mathcal{D}_v}, \pi_{\mathcal{D}_v}) \to \mathbb{R}.
$$

who receives the features, logits, and predictions of the validation set $\mathcal{D}_v$ and outputs a scalar value. When given a set of hyperparameters $\{\theta_1, \theta_2, \ldots, \theta_h\}$, and models trained with them as $\{f_{\theta_1}, f_{\theta_2}, \ldots, f_{\theta_h}\}$. We first extract the features, logits, and predictions of the validation set $\mathcal{D}_v$ using each model,

$$
\begin{aligned}
\phi_{\theta_i} &:= \{h(x_i)\}_{i=1}^{N^v}, \\
\ell_{\theta_i} &:= \{g(h(x_i))\}_{i=1}^{N^v}, \\
\pi_{\theta_i} &:= \{\arg\max g(h(x_i))\}_{i=1}^{N^v}, \forall i \in \{1, 2, \ldots, h\}.
\end{aligned}
$$

Then, given a set of validators $\{v_1, v_2, \ldots, v_n\}$, we evaluate the performance of each model on the validation set $\mathcal{D}_v$ using the validators. Specifically, given extracted features, logits, and predictions of the validation set $\mathcal{D}_v$ for each model as:

$\{(\phi_{\theta_1}, \ell_{\theta_1}, \pi_{\theta_1}), (\phi_{\theta_2}, \ell_{\theta_2}, \pi_{\theta_2}), \ldots, (\phi_{\theta_h}, \ell_{\theta_h}, \pi_{\theta_h})\}$, we calculate the performance of each model using each validator as,

$$
\begin{aligned}
&v_1(\phi_{\theta_1}, \ell_{\theta_1}, \pi_{\theta_1}), \\
&v_2(\phi_{\theta_2}, \ell_{\theta_2}, \pi_{\theta_2}), \\
&\qquad \ldots, \\
&v_n(\phi_{\theta_h}, \ell_{\theta_h}, \pi_{\theta_h}).
\end{aligned}
$$

Suppose we have $n$ validators and $h$ models. We denote the performance of each model on the validation set $\mathcal{D}_v$ using each validator as a matrix $M$:

$$M := \begin{bmatrix} v_1(\phi_{\theta_1}, \ell_{\theta_1}, \pi_{\theta_1}) & \cdots & v_n(\phi_{\theta_1}, \ell_{\theta_1}, \pi_{\theta_1}) \\ v_1(\phi_{\theta_2}, \ell_{\theta_2}, \pi_{\theta_2}) & \cdots & v_n(\phi_{\theta_2}, \ell_{\theta_2}, \pi_{\theta_2}) \\ \vdots & \ddots & \vdots \\ v_1(\phi_{\theta_h}, \ell_{\theta_h}, \pi_{\theta_h}) & \cdots & v_n(\phi_{\theta_h}, \ell_{\theta_h}, \pi_{\theta_h}) \end{bmatrix}.$$

Where each row represents the performance of a model under different validators. And each column represents the performance of different models under the same validator. Then across every column, we calculate the rank of each model based on the performance under the corresponding validator. We then get a rank matrix $R$:

$$R := \begin{bmatrix} r_{11} & r_{12} & \cdots & r_{1h} \\ r_{21} & r_{22} & \cdots & r_{2h} \\ \vdots & \vdots & \ddots & \vdots \\ r_{n1} & r_{n2} & \cdots & r_{nh} \end{bmatrix}.$$

Here each row represents the rank of a model under different validators. Then we can approximate the performance of each model by calculating the average rank across all validators. We denote the average rank of each model as a vector $A$:

$$A := \begin{bmatrix} \frac{1}{n}\sum_{j=1}^{n} r_{1j} \\ \frac{1}{n}\sum_{j=1}^{n} r_{2j} \\ \vdots \\ \frac{1}{n}\sum_{j=1}^{n} r_{hj} \end{bmatrix}.$$

To get the best performance model, we select the model with the lowest value in $A$. We put the validators we use and more detailed examples in appendix C.2.

## C.2 Validators

We deploy 7 semi-supervised learning validators, including

**RankMe Score [26]** We adapt RankMe Score to compute the feature matrix rank of the pre-trained data over both source and target data domains.

**Adjusted Mutual Information (AMI) [68]** We adapt AMI to compute the adjusted mutual information between predicted and cluster labels of the validation set.

**Adjusted Rand Index (ARI) [30]** ARI is a widely used metric in cluster analysis by measuring the similarity between two clustering solutions. Given $RI$ is the Rand Index between the true and predicted clustering, ARI is defined by:

$$ARI := \frac{RI - \text{Expected RI}}{\max(RI) - \text{Expected RI}}$$

**V-Measure [58]** Beside AMI, we also adapt V-Measure to measure the harmonic mean between homogeneity and completeness over the clustering labels and prediction.

**Fowlkes-Mallows Index (FMI) [24]** Measures the similarity between two clusterings. FMI is defined by:

$$FMI := \sqrt{\frac{TP}{TP + FP} \cdot \frac{TP}{TP + FN}}$$

**Calinski-Harabasz Index (CHI) [9]** CHI is an internal clustering measurement. Given $n$ sets of data point $\{x\}_1^n$ grouped into $k$ clusters $\{C\}_1^k$ with centroids $\{c\}_i^k$ and $\boldsymbol{c}$ is the overall centroid, CHI is defined by:

$$CHI := \frac{\left[\sum_{i=1}^{k} n_i ||c_i - c||^2\right] / (k-1)}{\left[\sum_{i=1}^{k} \sum_{x \in C_i} ||x - c_i||^2\right] (n-k)}$$

**Batch Nuclear-norm Minimization (BNM) [18]** An unsupervised domain adaptation method aims to generate predictions that favor both diversion and confidence; BNM maximizes the nuclear norm of the prediction matrix in a batch and repurposes it as a validation criterion.

### C.3 Hyperparameter Tuning Results

We present complete hyperparameter tuning results in Table 12.

## D Limitations

Our study primarily focuses on the image classification setting, following representative SSL frameworks such as FixMatch. While this scope allows for controlled and fair evaluation of SSL principles, it may limit the direct applicability of our findings to other tasks (e.g., detection or segmentation) that often demand task-specific architectures and training paradigms. Nevertheless, we believe the core insights from our analysis—such as the behaviors of pseudo-labeling, ensembling, and label-efficiency dynamics—are conceptually transferable. We leave a systematic exploration of these directions to future work.

| | | | DTD | | Sun397 | | Resisc45 | | Retinopathy | | Clevr-C | | KITTI | |
|---|---|---|---|---|---|---|---|---|---|---|---|---|---|---|
| | | | 3 | 6 | 3 | 6 | 1 | 2 | 40 | 80 | 10 | 20 | 5 | 10 |
| **CLIP** | | | | | | | | | | | | | | |
| LoRA | Oracle | Labeled-Only | 59.04% | 65.80% | 61.55% | 67.59% | 59.48% | 72.17% | 40.10% | 58.15% | 36.07% | 55.95% | 67.51% | 77.92% |
| | | Fixmatch | 56.81% | 68.56% | 62.17% | 72.78% | 51.41% | 79.51% | 35.59% | 44.09% | 37.11% | 49.53% | 59.77% | 61.88% |
| | | FlexMatch | 59.68% | 66.97% | 66.39% | 72.68% | 57.08% | 82.76% | 37.29% | 41.95% | 36.89% | 75.35% | 52.88% | 63.15% |
| | | SoftMatch | 59.79% | 68.03% | 66.31% | 70.87% | 78.70% | 83.65% | 42.05% | 43.50% | 36.30% | 71.20% | 50.35% | 65.12% |
| | | FineSSL | 63.14% | 69.10% | 56.25% | 61.78% | 74.17% | 84.73% | 35.03% | 31.58% | 33.55% | 40.26% | 54.85% | 51.48% |
| | RankMe | Labeled-Only | 52.45% | 62.18% | 58.29% | 57.04% | 55.71% | 59.73% | 30.27% | 48.43% | 35.21% | 44.01% | 60.06% | 64.28% |
| | | Fixmatch | 51.54% | 59.20% | 51.17% | 62.03% | 35.25% | 62.37% | 33.01% | 44.09% | 32.20% | 42.61% | 59.77% | 52.60% |
| | | FlexMatch | 59.68% | 66.33% | 65.36% | 52.01% | 51.24% | 82.76% | 31.82% | 39.90% | 30.94% | 27.98% | 44.59% | 55.41% |
| | | SoftMatch | 58.99% | 64.79% | 63.07% | 56.18% | 62.16% | 78.25% | 42.05% | 42.93% | 29.78% | 28.81% | 33.47% | 60.48% |
| | | FineSSL | 58.51% | 64.89% | 54.26% | 58.17% | 57.60% | 73.48% | 16.95% | 22.64% | 23.93% | 38.07% | 48.52% | 51.48% |
| | Random | Labeled-Only | 33.14% | 43.03% | 35.59% | 47.73% | 26.18% | 36.69% | 29.67% | 37.88% | 23.73% | 30.81% | 48.38% | 51.78% |
| | | Fixmatch | 54.34% | 64.57% | 58.07% | 68.34% | 44.74% | 73.10% | 33.68% | 41.39% | 30.48% | 40.82% | 48.95% | 52.46% |
| | | FlexMatch | 52.00% | 57.62% | 58.07% | 65.53% | 52.38% | 79.52% | 33.98% | 40.17% | 30.59% | 48.37% | 46.23% | 57.85% |
| | | SoftMatch | 53.56% | 58.12% | 60.11% | 64.99% | 67.42% | 81.32% | 36.93% | 42.32% | 30.21% | 49.76% | 42.24% | 59.21% |
| | | FineSSL | 46.61% | 51.06% | 52.82% | 57.12% | 47.60% | 80.56% | 26.31% | 28.22% | 29.29% | 37.48% | 52.37% | 50.26% |
| | Ours | Labeled-Only | 57.87% | 64.10% | 60.66% | 66.79% | 59.48% | 72.05% | 35.38% | 42.16% | 35.21% | 50.80% | 60.06% | 64.28% |
| | | Fixmatch | 56.81% | 68.56% | 62.17% | 72.78% | 47.56% | 77.41% | 32.45% | 39.09% | 32.20% | 42.61% | 59.77% | 52.60% |
| | | FlexMatch | 56.49% | 66.97% | 66.39% | 71.90% | 51.24% | 78.40% | 32.84% | 38.67% | 36.89% | 75.35% | 44.59% | 54.99% |
| | | SoftMatch | 59.79% | 68.03% | 66.31% | 70.87% | 78.70% | 83.65% | 42.05% | 42.93% | 29.78% | 71.20% | 42.90% | 60.48% |
| | | FineSSL | 63.14% | 69.10% | 56.25% | 61.78% | 74.17% | 83.48% | 35.03% | 30.42% | 33.55% | 38.07% | 53.73% | 48.24% |
| AdaptFormer | Oracle | Labeled-Only | 61.54% | 67.50% | 61.14% | 68.26% | 60.98% | 72.83% | 37.98% | 46.94% | 37.80% | 48.82% | 57.81% | 59.07% |
| | | Fixmatch | 61.22% | 67.50% | 64.63% | 71.19% | 63.06% | 80.49% | 38.88% | 46.37% | 34.27% | 46.41% | 55.13% | 60.34% |
| | | FlexMatch | 60.27% | 68.78% | 64.32% | 71.78% | 58.13% | 78.08% | 37.67% | 57.29% | 38.50% | 54.54% | 51.05% | 58.37% |
| | | SoftMatch | 61.33% | 69.31% | 65.33% | 70.17% | 65.81% | 82.67% | 38.16% | 42.66% | 36.66% | 56.08% | 47.12% | 57.24% |
| | | FineSSL | 61.91% | 68.14% | 61.97% | 67.83% | 73.95% | 84.17% | 26.55% | 34.22% | 25.97% | 36.87% | 54.15% | 52.60% |
| | RankMe | Labeled-Only | 56.86% | 65.43% | 61.14% | 67.35% | 52.98% | 70.43% | 31.02% | 39.28% | 30.43% | 41.95% | 55.13% | 56.68% |
| | | Fixmatch | 61.22% | 67.34% | 50.47% | 64.62% | 46.84% | 74.46% | 38.88% | 42.41% | 34.27% | 44.76% | 29.96% | 60.34% |
| | | FlexMatch | 59.95% | 67.13% | 38.34% | 49.39% | 50.10% | 73.56% | 37.67% | 38.30% | 38.50% | 43.43% | 34.18% | 54.15% |
| | | SoftMatch | 59.52% | 67.50% | 47.72% | 55.94% | 54.94% | 80.40% | 38.16% | 38.04% | 36.66% | 51.21% | 45.15% | 55.13% |
| 61.91 | | FineSSL | 61.91% | 68.14% | 58.37% | 64.86% | 73.95% | 84.17% | 25.27% | 29.78% | 25.97% | 36.87% | 42.19% | 47.12% |
| | Random | Labeled-Only | 31.40% | 41.17% | 37.04% | 47.87% | 22.73% | 34.31% | 21.31% | 26.43% | 20.56% | 26.57% | 44.75% | 45.08% |
| | | Fixmatch | 57.04% | 64.02% | 59.29% | 68.67% | 54.72% | 74.62% | 34.25% | 42.70% | 31.64% | 43.01% | 40.18% | 58.27% |
| | | FlexMatch | 51.91% | 58.12% | 55.59% | 63.96% | 55.30% | 75.92% | 34.53% | 43.84% | 33.32% | 50.06% | 43.08% | 55.98% |
| | | SoftMatch | 52.50% | 58.71% | 58.19% | 64.76% | 62.17% | 81.61% | 34.45% | 40.74% | 33.64% | 48.07% | 42.05% | 55.51% |
| | | FineSSL | 44.33% | 48.28% | 54.62% | 59.75% | 66.98% | 77.64% | 25.80% | 31.07% | 24.75% | 33.32% | 48.57% | 49.41% |
| | Ours | Labeled-Only | 61.54% | 64.95% | 61.14% | 68.23% | 60.98% | 72.83% | 34.38% | 38.99% | 37.80% | 48.63% | 57.81% | 59.07% |
| | | Fixmatch | 61.22% | 67.34% | 64.63% | 71.19% | 63.06% | 80.49% | 28.96% | 42.41% | 33.13% | 44.76% | 29.96% | 57.24% |
| | | FlexMatch | 60.27% | 68.78% | 64.32% | 70.73% | 58.13% | 78.08% | 31.02% | 35.93% | 38.50% | 52.21% | 34.18% | 54.15% |
| | | SoftMatch | 61.33% | 69.31% | 65.33% | 70.17% | 65.78% | 82.67% | 38.16% | 42.66% | 34.48% | 56.08% | 33.90% | 55.13% |
| | | FineSSL | 61.91% | 68.14% | 58.37% | 64.86% | 73.95% | 84.17% | 25.27% | 29.78% | 25.97% | 36.87% | 42.19% | 47.12% |
| **DinoV2** | | | | | | | | | | | | | | |
| LoRA | Oracle | Labeled-Only | 61.01% | 69.04% | 59.85% | 66.84% | 49.62% | 63.08% | 52.14% | 53.40% | 37.31% | 61.72% | 54.43% | 71.73% |
| | | Fixmatch | 57.39% | 68.40% | 58.72% | 71.38% | 36.60% | 66.65% | 38.11% | 44.67% | 35.65% | 48.18% | 37.27% | 48.80% |
| | | FlexMatch | 56.01% | 65.37% | 55.69% | 67.65% | 40.06% | 72.63% | 36.56% | 41.06% | 37.13% | 66.85% | 28.27% | 35.58% |
| | | SoftMatch | 53.62% | 65.59% | 60.41% | 67.34% | 54.79% | 81.46% | 38.64% | 44.48% | 36.59% | 65.07% | 27.43% | 43.60% |
| | | FineSSL | 66.97% | 72.66% | 56.14% | 62.07% | 69.78% | 87.38% | 45.23% | 28.80% | 27.60% | 34.03% | 46.95% | 49.65% |
| | RankMe | Labeled-Only | 46.76% | 63.56% | 55.34% | 60.97% | 42.22% | 43.33% | 33.35% | 47.85% | 31.75% | 39.69% | 46.69% | 56.40% |
| | | Fixmatch | 49.41% | 64.84% | 56.62% | 59.04% | 16.59% | 50.78% | 36.70% | 36.13% | 35.65% | 28.30% | 13.92% | 42.90% |
| | | FlexMatch | 56.01% | 63.83% | 55.69% | 50.05% | 33.32% | 72.14% | 35.86% | 28.86% | 32.84% | 33.20% | 28.27% | 33.61% |
| | | SoftMatch | 51.38% | 64.04% | 56.45% | 53.54% | 46.60% | 75.76% | 32.48% | 36.66% | 33.57% | 31.61% | 20.25% | 57.24% |
| | | FineSSL | 42.18% | 45.53% | 51.60% | 58.65% | 57.90% | 70.95% | 30.99% | 24.55% | 19.40% | 32.33% | 31.93% | 39.80% |
| | Random | Labeled-Only | 50.35% | 60.99% | 54.79% | 61.49% | 46.11% | 78.92% | 34.39% | 41.27% | 32.14% | 47.64% | 23.25% | 33.66% |
| | | Fixmatch | 54.06% | 64.18% | 55.65% | 67.00% | 23.78% | 56.07% | 36.01% | 40.66% | 31.43% | 39.16% | 21.99% | 38.07% |
| | | FlexMatch | 51.44% | 60.71% | 50.92% | 61.35% | 33.56% | 66.64% | 33.26% | 35.24% | 30.90% | 48.62% | 24.14% | 32.82% |
| | | SoftMatch | 50.35% | 60.99% | 54.79% | 61.49% | 46.11% | 78.92% | 34.39% | 41.27% | 32.14% | 47.64% | 23.25% | 33.66% |
| | | FineSSL | 58.30% | 63.28% | 54.12% | 60.55% | 45.20% | 79.85% | 34.20% | 24.00% | 23.92% | 32.70% | 39.10% | 44.49% |
| | Ours | Labeled-Only | 61.01% | 68.40% | 59.71% | 66.78% | 48.59% | 63.08% | 40.69% | 48.02% | 35.87% | 61.72% | 46.69% | 71.73% |
| | | Fixmatch | 57.39% | 68.40% | 56.62% | 71.38% | 36.60% | 66.65% | 38.11% | 41.16% | 35.65% | 40.99% | 14.77% | 42.90% |
| | | FlexMatch | 52.82% | 65.37% | 55.69% | 67.65% | 33.32% | 72.63% | 36.56% | 35.81% | 37.13% | 66.85% | 26.16% | 33.61% |
| | | SoftMatch | 53.62% | 64.04% | 60.41% | 67.34% | 46.60% | 79.54% | 32.05% | 44.48% | 33.57% | 65.07% | 27.43% | 27.85% |
| | | FineSSL | 66.97% | 72.66% | 54.62% | 62.07% | 69.78% | 87.38% | 30.99% | 24.55% | 24.77% | 31.73% | 31.93% | 39.80% |
| AdaptFormer | Oracle | Labeled-Only | 63.09% | 69.63% | 59.83% | 67.03% | 48.05% | 63.86% | 43.25% | 59.78% | 34.73% | 46.51% | 52.46% | 58.23% |
| | | Fixmatch | 59.47% | 70.37% | 60.41% | 69.68% | 27.56% | 62.30% | 38.73% | 45.55% | 32.65% | 39.25% | 33.33% | 50.35% |
| | | FlexMatch | 57.07% | 65.90% | 56.01% | 67.53% | 51.29% | 64.02% | 33.87% | 42.18% | 33.35% | 47.67% | 34.32% | 40.51% |
| | | SoftMatch | 58.67% | 66.17% | 59.97% | 66.65% | 53.06% | 78.06% | 34.65% | 40.80% | 33.37% | 46.55% | 25.60% | 37.97% |
| | | FineSSL | 66.38% | 71.17% | 56.17% | 63.34% | 62.38% | 70.19% | 28.75% | 31.61% | 25.79% | 30.03% | 31.79% | 48.52% |
| | RankMe | Labeled-Only | 58.51% | 4.73% | 56.57% | 60.78% | 44.71% | 54.22% | 35.82% | 37.82% | 29.67% | 40.65% | 52.32% | 55.41% |
| | | Fixmatch | 51.65% | 66.28% | 45.86% | 59.07% | 27.56% | 58.54% | 38.73% | 45.55% | 26.99% | 29.78% | 19.83% | 50.35% |
| | | FlexMatch | 52.13% | 62.34% | 41.94% | 51.05% | 45.79% | 62.10% | 27.50% | 33.58% | 25.49% | 27.55% | 34.32% | 27.85% |
| | | SoftMatch | 47.71% | 61.12% | 43.25% | 53.20% | 49.71% | 70.83% | 30.02% | 40.34% | 26.05% | 28.76% | 23.91% | 29.54% |
| | | FineSSL | 42.87% | 45.32% | 53.55% | 60.31% | 62.32% | 69.73% | 17.66% | 23.04% | 25.79% | 28.73% | 31.65% | 46.41% |
| | Random | Labeled-Only | 34.69% | 45.25% | 38.53% | 49.57% | 20.67% | 29.72% | 29.12% | 35.31% | 19.95% | 25.34% | 42.13% | 45.34% |
| | | Fixmatch | 55.55% | 65.37% | 54.71% | 65.96% | 23.97% | 58.65% | 35.31% | 42.21% | 29.48% | 35.97% | 23.58% | 34.74% |
| | | FlexMatch | 53.10% | 61.76% | 51.12% | 61.99% | 43.72% | 60.70% | 31.60% | 38.66% | 30.46% | 39.13% | 28.50% | 31.60% |
| | | SoftMatch | 52.34% | 61.68% | 52.93% | 60.77% | 51.27% | 74.91% | 32.71% | 37.77% | 30.78% | 40.10% | 24.66% | 32.82% |
| | | FineSSL | 57.77% | 62.41% | 55.23% | 61.34% | 60.01% | 67.41% | 24.13% | 26.32% | 25.39% | 29.44% | 31.22% | 46.65% |
| | Ours | Labeled-Only | 62.98% | 67.61% | 58.87% | 67.03% | 47.00% | 57.41% | 34.42% | 53.07% | 27.33% | 41.79% | 52.32% | 49.79% |
| | | Fixmatch | 59.47% | 70.37% | 57.85% | 69.68% | 26.41% | 62.30% | 38.73% | 45.55% | 32.65% | 38.89% | 19.83% | 50.35% |
| | | FlexMatch | 57.07% | 65.90% | 56.01% | 67.53% | 51.29% | 64.02% | 33.43% | 40.20% | 33.35% | 42.16% | 34.32% | 27.85% |
| | | SoftMatch | 58.67% | 66.17% | 59.97% | 66.65% | 53.06% | 78.06% | 33.47% | 32.17% | 32.91% | 45.00% | 24.47% | 30.94% |
| | | FineSSL | 64.04% | 71.17% | 55.96% | 63.34% | 62.32% | 69.73% | 17.66% | 24.30% | 25.79% | 30.03% | 31.79% | 45.01% |

Table 12: Complete hyperparameter tuning results for all datasets, models, and SSL baselines. For single validators we only include RankMe.

