# OpenReview forum: "Revisiting Semi-Supervised Learning in the Era of Foundation Models"
_NeurIPS.cc/2025/Conference — NeurIPS 2025 poster_

### Official Review · Reviewer_BzLX · 2025-06-23

**Clarity:** 4
**Significance:** 3
**Originality:** 2
**Rating:** 5
**Confidence:** 4

**Summary:**

Foundation models are often used as backbones for downstream tasks, like classification. When applied to bespoke classification datasets, baseline performance may not be as high as users require. In these settings, it is common that users have a small set of labelled data and much more unlabelled data, making it suitable for semi-supervised learning.

The authors construct an evaluation dataset with classification tasks in which existing foundation models (DINOv2 and CLIP) perform poorly (30-70% accuracy).

The authors explore if existing semi-supervised learning methods (FixMatch, FlexMatch, SoftMatch) can be used to fine-tune foundation models to improve their performance on these bespoke datasets. When fine-tuning the models, the authors test both full parameter fine-tuning and parameter-efficient fine-tuning (LORA and Adaptformer).

The authors find that:
- the simple approach, full parameter fine-tuning on just a few labelled data samples is very effective and existing SSL methods do **not** confer an advantage.
- parameter-efficient fine-tuning improves performance of all training methods, again providing no advantage to SSL methods which make use of extra unlabelled data.

The authors propose a new method for SSL on foundation models that uses parameter efficient self-training and ensembling to improve model performance. They use a small amount of labelled data to fine-tune the model and use the resulting model to generate pseudo-labels for self-training. They ensemble several base models and parameter-efficient fine-tuning methods. They try three different supervisory signals leveraging the ensembling: (Mean Logits, Mean Probs, and Mean Labels). They empirically demonstrate that their approach with Mean Labels performs the best on their evaluation dataset.

**Questions:**

None

**Ethical Concerns:**

["NO or VERY MINOR ethics concerns only"]

**Final Justification:**

I have read the other reviews and their responses and I have no new concerns. I intend to maintain my rating (5) as the significance, clarity, and soundness of the work are strong, but the methodological novelty remains limited.

**Limitations:**

The authors test on classification and on 6 datasets, this is good evaluation benchmark but not comprehensive. It's addressed to some degree in the text, but not in a specific section independently.

**Paper Formatting Concerns:**

Mentioned in minor weaknesses

**Quality:**

4

**Strengths And Weaknesses:**

Strengths:
- The problem choice is important. Many users try to apply foundation models to specific problems and find mixed results. Establishing a benchmark and methods for this specific setting is worthwhile.
- The paper is written clearly. It is easy to follow the investigation of prior work and the intuition behind their proposed approach.
- The authors ablate various components of their algorithm to precisely understand the contributions of each component. They also explore several options for each step. They choose recent methods for their baseliens and conduct hyperparameter sweeps to try and achieve a fair comparison.
-  The method they are proposing is a novel combination of established methodologies.

Weaknesses:
- While the evaluation benchmark is reasonable, it is still relatively limited in comparison to the diversity of tasks users may be interested in.
- The methodology is fairly similar to the one proposed in FineSSL. The authors work does introduce two additional contributions: (1) more breadth in the evaluation tasks, backbones and baselines explored and (2) the idea to ensemble different methods to achieve better performance.

Minor:
- L113 - "first" comprehensive study of SSL in the era of foundation models is a broad claim. A more specific claim would be more appropriate.
- L187 - "few" is ambiguous, refer to Table 2.
- L240 - would be better to keep section header with the text it refers to.
- L867-869 - potentially capitalize appendix?

Justification: I recommend a 5 because the problem is important, the paper is technically sound, and presented well. I do not believe the results or method is ground-breaking, so I cannot recommend a higher score.

---

> ### Author Rebuttal · Authors · 2025-07-30
>
> Thank you for your support of our work and your constructive feedback.
>
> **Concern 1: Selection of Evaluation Benchmark**
>
> We appreciate this observation and thank you for pointing out the limitation regarding task diversity in our evaluation benchmark.
> (1) Regarding the dataset in our benchmark: ​​While our study centers on classification, we deliberately ensured diversity by following the VTAB sub-categories (*NATURAL*, *SPECIALIZED*, and *STRUCTURED*), selecting at least two representative datasets from each. This design balances task diversity, domain coverage, and practical feasibility. To our knowledge, such a structured evaluation within the context of SSL remains underexplored. We hope our benchmark helps fill this gap and serves as a solid foundation to inspire future research on semi-supervised learning with VFMs across a broader range of tasks beyond classification.
>
> (2) Regarding the task type of our benchmark: We agree that extending to tasks beyond classification is valuable. However, such tasks often involve task-specific architectures and training paradigms. To maintain focus on core machine learning challenges, we align with prior SSL works (e.g., FixMatch) by prioritizing image classification. We believe the core SSL principles and insights from our study are transferable. For instance, UniMatch [A], a popular SSL segmentation method, was directly inspired by FixMatch, and many SSL segmentation approaches [B, C] adopt self-training and pseudo-labeling. We appreciate the suggestion and will discuss this limitation in our revised manuscript.
>
> **Concern 2: Comparing to FineSSL**
>
> Thank you for your thoughtful observation. We agree that our work shares a similar high-level methodology with FineSSL [D] – both adopt pseudo-labeling in conjunction with PEFT to improve sample efficiency in semi-supervised learning with VFMs. FineSSL presents a simple and well-structured approach, and our work also provides valuable extension along two key axes: (1) building a more systematic and diverse benchmark across datasets, backbones, and PEFT methods, and (2) exploring model diversity through ensembling to further enhance pseudo-label quality.
>
> We see FineSSL as a timely and valuable contribution to this emerging research direction. Our aim is to build upon such efforts to establish a strong and extensible baseline for future work. We will revise the related work and discussion sections to better clarify the connection and distinction between our method and FineSSL.
>
> **Concern 3: Scope of the Work**
>
> Thank you for your positive evaluation regarding the importance of the problem, technical soundness, and clarity of presentation. We also fully respect your assessment regarding the level of novelty. While the current results may not appear ground-breaking in isolation, we believe the proposed framework and benchmark introduce a new perspective in the SSL community that would likely lead to broader implications. We see this as a step toward a new paradigm that could catalyze future research and lead to impactful applications within the community.
>
> **Concern 4: Section for Limitations**
>
> Thank you for the constructive suggestion. We agree that, while we briefly acknowledge certain limitations in the main text (e.g., the focus on classification tasks [line 75] and the selection of six datasets [line 135]), these points would be clearer and more transparent if presented in a dedicated section. In the revised version, we will add an explicit "Limitations" section to clearly outline the scope of our current study and discuss directions for broader task coverage in future work.
>
> **Concern 5: Formatting issues**
>
> Thank you for pointing out our flaws in the formatting. We’ll revise them in the future manuscript and ensure a clean and professional final version.
>
> **References**
>
> [A] Revisiting weak-to-strong consistency in semi-supervised semantic segmentation. CVPR’23.
>
> [B] St++: Make self-training work better for semi-supervised semantic segmentation. CVPR’22.
>
> [C] Re-distributing biased pseudo labels for semi-supervised semantic segmentation: A baseline investigation. ICCV21.
>
> [D] Erasing the Bias: Fine-Tuning Foundation Models for Semi-Supervised Learning. ICML’24.

---

> > ### Comment · Reviewer_BzLX · 2025-07-31
> >
> > Thank you for your response. I have read the other reviews and their responses and I have no new concerns. I intend to maintain my rating (5) as the significance, clarity, and soundness of the work are strong, but the methodological novelty remains limited.

---

> > > ### Author Response · Authors · 2025-08-07
> > > **Thank you for your positive feedback**
> > >
> > > Dear Reviewer BzLX,
> > >
> > > Thank you for your prompt feedback. We are glad to know that you have no new concerns and remain positive and supportive of our paper regarding its significance, clarity, and soundness. We will incorporate our rebuttal into the final version.

---

### Official Review · Reviewer_gEWg · 2025-06-30

**Clarity:** 3
**Significance:** 2
**Originality:** 2
**Rating:** 3
**Confidence:** 4

**Summary:**

This paper studies the problem Semi-supervised Learning (SSL) in the context of foundation models, and found that standard SSL technique only incrementally benefits foundation models.To make unlabelled data pay off, the authors introduce V-PET—a one-shot self-training routine that simply averages the one-hot predictions of several differently fine-tuned backbones and then retrains on that soft target—showing consistent gains over prior SSL baselines on a new, harder VTAB-derived benchmark.

**Questions:**

1. I think another informative point authors could try to add is how foundation models' performances sclaes as we increase the number/proportion of the labeled data, this is expected to behave somewhat differently to standard SSL classifiers as they perhaps need more labeled data since they are trained from the scratch.

2. Are foundation models more vulnerable to the problem of imbalance comparing to classifical SSL classifiers?

**Ethical Concerns:**

["NO or VERY MINOR ethics concerns only"]

**Final Justification:**

After a through discussion with the authors, while most of my concerns are somewhat addressed, it seems that when authors reporting the computational efficiency, their original experiments lacks rigor and soundness, making the computational overhead seemingly smaller (see authors' last two responses for details), therefore my concern remain for this aspect.

In addition, the technical novelty of this paper remains limited in my view. Therefore, I tend to maintain my original evaluation as leaning towards rejection.

**Limitations:**

N.A.

**Paper Formatting Concerns:**

N.A.

**Quality:**

3

**Strengths And Weaknesses:**

**Strengths:**

1. The problem being studied in this paper is very crucial and of high interest to the SSL community - what role does SSL play in the era of foundation models, and what sets it apart from classical SSL?

2. The take-away message is informative and perhaps stirs further thinking in this domain - naively utilizing unlabeled data does not seems to contribute much for performance gain.

3. The tested cases covered in the experiment are rather comprehensive and informative.

4. This paper is overall well-written and easy to follow.

**Weaknesses:**

1. One major weakness of this paper appears to be the computational complexity of the proposed method; this method requires an ensemble of multiple foundation models, and considering the complexity of the foundation models, the computational cost of the proposed method seems significant. Moreover, efficiency studies are missing in this paper.

2. More ablation studies are needed to further highlight the importance of self-training; authors mainly proposed two techniques, self-training and ensemble, and combining them together witness performance gain. However, such performance gain could solely stem from ensemble rather than self-training.

3. The main insights from this paper are not significant enough - the resulting methodology essentially stems from ensemble and self-training, and all of these are not very relevant to the "unique properties of foundation models."

4. The coverage of techniques and tricks is not comprehensive enough; some other very important aspects, such as threshold of pseudo-label (FlexMatch, AdaMatch), data-augmentation and consistency training (FixMatch), exponential moving average, label alignment, and other techniques to counter training imbalance, are not discussed or mentioned. However, researchers in SSL might be interested in learning why or why those tricks still work for foundation models.

---

> ### Author Rebuttal · Authors · 2025-07-28
>
> Thank you for your interest in our work and for your constructive feedback. Meanwhile, we respectfully think some of the concerns may result from misunderstandings, and we apologize if these were caused by the organization of our paper. Before addressing your specific concerns, we would like to provide a brief overview of our paper and algorithm.
>
> In the conventional setting of **few-shot semi-supervised learning**, our method follows a **4-stage pipeline**:
>
> 1. Supervised Parameter-Efficient Fine-Tuning: We select a set of either *(a) Different PEFT methods (PET)* or *(b) Different VFMs and PEFT methods (V-PET)*, and fine-tune them using **only a few labeled samples**.
> 2. Pseudo-Label Generation: The fine-tuned model is then used to generate pseudo-labels for the unlabeled data through **single forward pass**.
> 3. Pseudo-Label Ensemble: Multiple sets of pseudo-labels are ensembled through our *mean-label* strategy.
> 4. Ensemble Training: The final VFM is trained on the originally unlabeled data, using the ensemble pseudo-labels as supervision.
>
> Based on the workflow. We’d like to address your concerns one by one as follows:
>
> **Concern 1: Computational Complexity**
>
> We acknowledge that, like other ensemble approaches, our method introduces some additional computational cost overhead. However, thanks to **the efficiency of stage 1 and stage 2** listed above, this extra cost remains negligible. To better illustrate this trade-off, we summarize the end-to-end computation time in the table below:
>
> |           | ~Sec/Epoch | Epochs | ~Time (Mins) |
> |-----------|------------|--------|--------------|
> | FixMatch  | 93         | 30     | 46.5         |
> | FlexMatch | 93         | 30     | 46.5         |
> | SoftMatch | 93         | 30     | 46.5         |
> | FineSSL   | 93         | 30     | 46.5         |
>
> |       | ~Sec/Epoch (SFT) | Epochs (SFT) | #SFT Models | ~Sec/Epoch (Ensemble) | Epochs (Ensemble) | ~Total Time (Mins) |
> |-------|------------------|--------------|-------------|------------------------|-------------------|---------------------|
> | PET   | 2                | 50           | 2           | 93                     | 30                | 49.8                |
> | V-PET | 2                | 50           | 4           | 93                     | 30                | 53.1                |
>
> As shown, PET incurs only ~3.3 additional minutes compared to baselines (1.07×), and V-PET ~6.6 minutes (1.14×). We believe this is a reasonable overhead given the performance gains achieved.
>
> **Concern 2: Self-Training vs. Ensemble**
>
> We appreciate your question. Before we discuss ensemble and self-training, we want to first clarify that our contributions extend beyond simply employing these techniques, but also lie in the insight that PEFT and multiple VFMs can generate high-quality, complementary pseudo-labels that make ensemble learning effective.
>
> Regarding the performance gain, both ensemble and self-training are essential components. Self-training, as a SSL method, provides a principled way to leverage unlabeled data by generating pseudo-labels and fine-tuning the model in a supervised manner. Ensemble, on the other hand, serves to enhance the quality of these pseudo-labels. It is important to note that self-training is indispensable in our framework — without it, ensemble alone cannot fine-tune the pre-trained VFM.
> In our main results (Table 3), ST, PET, and V-PET all incorporate self-training. The distinction lies in how pseudo-labels are generated. We observe that ST alone does not consistently outperform the labeled-only baseline, indicating that the unlabeled data have not been effectively leveraged—likely due to the low quality of the pseudo-labels. In contrast, our approaches—PET and V-PET—use pseudo-label ensembles from multiple VFMs and PEFTs, enabling self-training to yield significant improvements over the labeled-only setting. This highlights the critical role of pseudo-label quality and underscores the effectiveness of our approach.
>
> **Concern 3: Limited Novelty and Relevance to Foundation Models**
>
> We respectfully disagree with the view that our methodology does not leverage the unique properties of VFMs. As mentioned in the paper [lines 51 - 56], our approach is built upon two core properties of VFMs. First, their ability to adapt quickly with minimal supervision — i.e., few-shot generalization. Second, the difference in their pretraining distribution or objectives. Our method explicitly exploits these capabilities by using VFMs to generate high-quality and complementary pseudo-labels from a few labeled examples, which enables us to further learn from unlabeled data.
>
> While the use of ensemble and self-training may appear conventional, our key contribution lies in showing how these techniques can be adapted to the VFM regime — not as arbitrary choices, but as mechanisms that align with VFMs’ unique strengths (e.g., stable zero-shot predictions, diverse representations). Moreover, by establishing a controlled benchmark and introducing a simple yet effective baseline, we aim to lay the groundwork for a principled exploration of SSL in the context of VFMs. We believe this is both timely and impactful, given the growing interest in adapting large-scale models under limited supervision.
>
>
> **Concern 4: Incorporating Other SSL Techniques**
>
> Thank you for your question. We acknowledge that there are many existing SSL techniques. While thoroughly revisiting these tricks would be valuable to the community, we humbly believe this is orthogonal and beyond our focus. Indeed, as highlighted in paper [lines 34, 67], one of our aims is to establish a simple SSL pipeline—ideally with as few tricks as possible — by appropriately leveraging the unique properties of VFMs and PEFT.
>
> That said, we do think that the existing SSL tricks and techniques (e.g., data augmentation, self-consistency) have the potential to further improve our method.
>
> To explore this direction, we conducted additional experiments to evaluate the compatibility of our method with both weak and strong augmentations and self-consistency techniques. In the experiments, we fixed the hyperparameter settings under the **DTD-3SHOT, DINOv2** configuration, using **LoRA** for ensemble training. We evaluated the performance of our method in combination with existing SSL techniques under two main settings:
>
> 1) **Self-Consistency**: Pseudo-labels are generated from weakly augmented (random horizontal flip) unlabeled data and averaged over **10** stochastic forward passes.
>
> 2) **Strong Data Augmentation**: During ensemble training, we add a cross-entropy loss term computed on RandAugment(m=3, n=5)-augmented samples.
>
> |                     | **w/o Self-Consistency** | **+ Self-Consistency** |
> |---------------------|------------------------|----------------------|
> | **w/o Strong Aug**  | 63.62                | 63.35              |
> | **w/ Strong Aug**           | 64.26                | 64.26              |
>
> We observe that strong augmentation yields modest performance gains, whereas self-consistency offers no additional benefit. This suggests that the effectiveness of such techniques may be less pronounced in our method or requires further tuning. We consider a more comprehensive exploration of such combinations as a promising direction for future work.
>
>
> **Question 1: Foundation Models’ Scaling Law for Labeled Data**
>
> Thank you for your constructive suggestion. Compared to training standard SSL classifiers from scratch, foundation models indeed require less labeled data – that is precisely why existing works leveraging foundation models for SSL [B][C], including us, primarily focus on low-shot settings. That said, we agree that systematically studying this question is an interesting direction. In response, we have extended our supervised fine-tuning part of **Table 3** on the *DTD* dataset from **3-SHOT** to **48-SHOT** setting. The updated results are shown below,
>
>
> |                    | DTD N3 | DTD N6 | DTD N12 | DTD N24 | DTD N48 |
> |--------------------|--------|--------|---------|---------|---------|
> | DINOv2 LoRA        | 61.01 | 68.40 |  74.26 |  78.19 |  82.45 |
> | DINOv2 AdaptFormer | 62.98 | 67.61 |  74.52 |  80.48 |  83.03 |
> | CLIP LoRA          | 57.87 | 64.10 |  69.84 |  74.57 |  78.46 |
> | CLIP AdaptFormer   | 61.54 | 64.95 |  71.76 |  76.17 |  79.36 |
>
> As we can see, performance on low-shot settings (**N3, N6**) is already strong, thanks to the rich priors encoded in foundation models. As the number of labeled examples increases, performance continues to improve but with diminishing returns – indicating a flatter scaling curve compared to standard SSL classifiers trained from scratch. We appreciate the reviewer for highlighting this important difference, and we’ll revise the final version to have this study.
>
> **Question 2: Foundation Models’ Vulnerability to Imbalanceness**
>
> Thank you for the question. While we do not explicitly explore this question in the main paper, we respectfully believe that using foundation models as the backbone would make the overall SSL training more robust to class imbalance, compared to training from scratch (in conventional SSL methods). This is because foundation models often enable few-shot adaptation, which prevents minor categories from being simply dominated by major categories. Notably, our V-PET approach is a simple pseudo-labeling baseline without any rebalancing techniques, yet it still performs competitively. This suggests a certain degree of inherent robustness to imbalance in foundation models. We consider this a promising direction for future exploration.
>
> **References**
>
> [A] Lessons and insights from a unifying study of parameter-efficient fine-tuning (PEFT) in visual recognition. CVPR’25.
>
> [B] Erasing the Bias: Fine-Tuning Foundation Models for Semi-Supervised Learning. ICML’24.
>
> [C] USB: A Unified Semi-supervised Learning Benchmark for Classification. NeurIPS’22.

---

> > ### Comment · Reviewer_gEWg · 2025-08-05
> >
> > Thank the authors for their quality response, here are some remaining concerns:
> >
> > > W1:
> >
> > Thanks for providing these results, however, based on my experiences in this domain, it seems that authors implementation might be simplified? For example, the training time for FixMatch, FlexMatch, SoftMatch are exactly the same, while I understand they are similar in a fundamental way, it seems uncommon their training time are precisely the same.
> >
> > Also, the training time of Ensemble is also precisely 93 seconds per epoch? This means when you're doing the ensemble you are not increasing additional compute at all compare to baselines? Can authors elaborate on this point to clarify this?
> >
> > > W2:
> >
> > "without it, ensemble alone cannot fine-tune the pre-trained VFM". Well, you still can use labeled data only to fine-tune and apply ensemble at inference time for the predictions - based on my experience this usually works quite well. The reason I ask this is because I want to know how much VFMs benefits from self-training exactly and disentangle ensemble from it can help us understand this perspective.

---

> > > ### Author Response · Authors · 2025-08-06
> > >
> > > **Response for W1**
> > >
> > > Thank you for raising this concern.
> > >
> > > 1. Regarding the reported training time: it is an *approximate estimation* based on the number of epochs multiplied by a uniform per-epoch duration (e.g., 93 seconds), measured from representative runs. This approximation is used across all baselines—including *FixMatch*, *FlexMatch*, *SoftMatch*, *FineSSL*, and our *V-PET* to provide readers with an intuitive sense of relative training cost. In practice, these methods exhibit similar per-epoch training times under the same backbone, as their additional operations — such as thresholding or alignment — incur negligible overhead compared to the main loop. We agree that this simplification could be clarified, and we apologize for any confusion it may have caused.
> > >
> > > 2. Regarding the ensemble, we note that it *does not* involve training multiple branches simultaneously. All ensemble components are trained independently beforehand (*Stage 1* as described above). During ensembling, we simply perform a forward pass through each pre-trained model to generate pseudo-labels (*Stage 2*), and then train the final model using these labels following standard pseudo-labeling procedures (*Stage 3*).
> > >
> > > **Response for W2**
> > >
> > > Thank you for your insightful comment.
> > >
> > > We agree that using labeled data alone to fine-tune a pre-trained VFM and applying ensemble at inference time is a valid and often effective strategy. However, our goal is to explore **semi-supervised learning (SSL)**—specifically, how to leverage unlabeled data to improve model performance and ultimately produce a single deployable model.
> > >
> > > To better understand how much benefit VFMs can gain from self-training beyond ensembling, we conducted additional analysis under the *DTD 3-shot* setting. Specifically, we compared the accuracy of pseudo labels before and after the second-stage training. The results are summarized below:
> > >
> > > | Backbone | Method              | PL Accu (Before) | PL Accu (After) (Δ) | Test Accu |
> > > |----------|---------------------|------------------|----------------------|-----------|
> > > | CLIP     | LoRA / PET          | 61.7             | 65.2 (+3.5)          | 63.4      |
> > > | CLIP     | LoRA / V-PET        | 65.7             | 67.2 (+1.5)          | 65.6      |
> > > | CLIP     | AdaptFormer / PET   | 61.7             | 65.6 (+3.9)          | 63.8      |
> > > | CLIP     | AdaptFormer / V-PET | 65.7             | 67.3 (+1.6)          | 65.7      |
> > > | DINOv2   | LoRA / PET          | 62.7             | 67.3 (+4.6)          | 66.7      |
> > > | DINOv2   | LoRA / V-PET        | 65.7             | 68.2 (+2.5)          | 67.8      |
> > > | DINOv2   | AdaptFormer / PET   | 62.7             | 67.8 (+5.1)          | 67.2      |
> > > | DINOv2   | AdaptFormer / V-PET | 65.7             | 68.3 (+2.6)          | 68.2      |
> > >
> > > Across all backbones and methods, we consistently observe improvements in pseudo-label accuracy (positive Δ) after the second-stage training. Indicating that self-training contributes additional learning signal beyond ensembling alone.
> > >
> > > We hope this helps clarify the unique role of self-training in our framework, and we appreciate your suggestion in encouraging us to quantify its contribution more explicitly.

---

> > > > ### Comment · Reviewer_gEWg · 2025-08-08
> > > >
> > > > I thank the authors for their prompt response, however I still have question regarding to the efficiency of this approach:
> > > >
> > > > > W1:
> > > >
> > > > I understand your pipeline completely, what I am asking is why is there no computational overhead when you performs forward passing over multiple pre-trained VFMs?

---

> ### Author Response · Authors · 2025-08-09
>
> **Re-Response: About Computation Efficiency**
>
> Thank you for your comment. After carefully reading your question again, we realize that our prior response did not fully answer it. We apologize for it, and we provide a more detailed response as follows.
>
> To more easily decompose and analyze the computational time, please allow us to review our overall pipeline:
>
> Supervised Parameter-Efficient Fine-Tuning: We select a set of (a) Different PEFT methods (PET) or (b) Different VFMs and PEFT methods (V-PET), and fine-tune each of them using only a few labeled samples per class.
> Pseudo-Label Generation: Multiple fine-tuned models are then used to generate pseudo-labels for each unlabeled data sample.
> Pseudo-Label Ensemble: Multiple pseudo-labels for each unlabeled data sample are ensembled through our mean-label strategy.
> Ensemble Training: The final VFM (one backbone) is trained (using one PEFT method) on the abundant unlabeled data, using the **already pre-computed ensembled pseudo-labels** in stage 3 as supervision.
>
> Within this pipeline, the multiple VFMs (and multiple PEFT methods) are involved only in the first three stages. In stage 4, only one VFM and PEFT method is considered and trained, such as CLIP-LoRA or DINOv2-AdaptFormer, as reported in Table 3 of the main paper.
>
> At first glance, it seems that the first three stages are where the computational bottleneck is, since they involve multiple VFMs. However, in reality, stage 4 — which involves one VFM and PEFT — indeed contributes to the major portion of computational time. This is because stage 4 involves training over all the unlabeled data (e.g. SUN397-3SHOT are 68,410 samples). In contrast, stage 1 only involves training over a few labeled data samples (same dataset, only 397*3=1,191 samples). As such, **the computational time is relatively short compared to stage 4**, even if we train multiple VFM-PEFT pairs. In stages 2 and 3, where we apply multiple pre-trained VFM-PEFT pairs on all the unlabeled data for pseudo-label ensembling, we emphasize that each VFM-PEFT pair **only needs to perform one single forward pass for each unlabeled sample**.
>
> In our original rebuttal, *~Sec/Epoch (SFT) refers to stage 1 for each VFM-PEFT pair; ~Sec/Epoch (Ensemble) refers to stage 4*; *~Total Time (Mins) refers to the total time of the four stages*. We apologize for confusing as we did not provide further explanation for the naming. Note that, since in stage 4, we only train one VFM-PEFT pair — the ensemble pseudo-labels have been pre-computed in stage 3 and remain fixed — the ~Sec/Epoch will be approximately the same as other baseline methods.
>
> **Response: Why is there no computational overhead when you perform forward passing over multiple pre-trained VFMs?**
>
> We appreciate your question. There is indeed computational overhead when we train and infer with multiple pre-trained models. This was partially reflected in the ~Total Time (Mins) column (in the original rebuttal): V-PET takes a longer time than PET. In the original table, we **neglect the computational overhead in stages 2 and 3 since it is faster than stages 1 and 4**. However, we realize that for a fair comparison to the baselines, this should have been included. In the following, we **refine the table**, including computational time from all the stages.
>
> ||Stage1||||Stage2&3|||Stage4|||Total|
> |--------|----------------------|--------|----------|---------------------|-----------------------------|----------|---------------------|-------------|--------|---------------------|--------------------|
> |Method|~Sec/Epoch|Epochs|#Models|~Time(Mins)|~Sec/ForwardPass|#Models|~Time(Mins)|~Sec/Epoch|Epochs|~Time(Mins)|~Time(Mins)|
> |PET|2|50|2|3.3|15|2|0.5|93|30|46.5|50.3|
> |V-PET|2|50|4|6.6|15|4|1.0|93|30|46.5|54.1|
>
> We hope this detailed table addresses your concern. We note that this makes our computation time **1.08x for PET** and **1.16x for V-PET** over the baselines, respectively. That said, regarding your original concern in the review, “the computational cost of the proposed method seems significant,” we respectfully think a 1.16x computational time does not introduce a significant overhead. This is because the computational overhead of multiple VFMs only appears in stages 1-3, while stage 4 remains the major computational bottleneck. This also implies that our total computational time **does not simply scale linearly** when more VFMs and PEFT are involved. Specifically, considering Figure 5 in the main paper, where we study six VFMs, the overall computational time would go to ~58 minutes, roughly 1.24x over baselines instead of 6x.
>
> Once again, we apologize for any potential confusion.

---

### Official Review · Reviewer_DJbk · 2025-07-02

**Clarity:** 3
**Significance:** 2
**Originality:** 2
**Rating:** 4
**Confidence:** 4

**Summary:**

This paper investigates the applicability of semi-supervised learning (SSL) to vision foundation models (VFMs), specifically CLIP and DINOv2. The paper systematically evaluates whether existing SSL methods remain effective when applied to VFMs. Based on findings, the paper proposes V-PET, a simple yet effective self-training SSL method that ensembles pseudo-labels generated across diverse parameter-efficient fine-tuning (PEFT) methods and VFM backbones.

**Questions:**

1. Beyond empirical observations, could you explain more clearly why ensemble pseudo-labeling works better than using individual PEFT pseudo-labels in VFM settings? While this is discussed in Figure 1 and lines 255–263, a clearer reason would strengthen the claims.

2. In Table 3, what backbones and PEFT methods were used for the existing methods? Please clarify. If the same backbones and PEFT methods were used, your method’s training cost would increase with the number of PEFT methods and VFMs used. In that case, are there any additional benefits beyond accuracy, such as improved training stability? If different backbones or PEFT methods were used, reporting training time or FLOPs would help highlight the advantages of your approach.

3. In Table 3, the proposed methods seem to show slightly lower performance on only some datasets (e.g., Resisc45 and Retinopathy). Could you explain why this happens?

**Ethical Concerns:**

["NO or VERY MINOR ethics concerns only"]

**Final Justification:**

The authors have provided a detailed and thoughtful rebuttal that successfully addresses the main concerns raised in the initial review. In particular, their explanation of why ensemble pseudo-labeling is effective is well-reasoned and supported by additional empirical evidence.

I maintain my original score, but I am now more confident in my assessment. I hope the authors incorporate the discussed points to further improve the clarity and completeness of the final version.

**Limitations:**

It would be helpful to explicitly discuss the limitations of the proposed method, particularly regarding its applicability beyond classification tasks.

**Paper Formatting Concerns:**

.

**Quality:**

3

**Strengths And Weaknesses:**

This paper provides a thorough empirical study with extensive evaluations. The structure of the paper is clear, and it offers insights into SSL in the era of foundation models. While the proposed method is effective despite its conceptual simplicity, the paper offers limited theoretical insights into why ensemble pseudo-labeling is particularly effective in VFM settings.

---

> ### Author Rebuttal · Authors · 2025-07-28
>
> Thank you for your interest in our work, and we greatly appreciate your valuable feedback. We would like to address all your concerns, point by point, as follows:
>
> **Concern 1: Why Ensembling Works Better**
>
> Thank you for raising this insightful question. We would like to reiterate and further elaborate on why ensemble pseudo-labeling is particularly beneficial in the VFM+PEFT setting, building on the evidence already discussed in Section 5.2.
>
> Unlike traditional SSL methods that ensemble models trained via random initialization, our method leverages the inherent diversity across different PEFT strategies and VFM pretraining objectives. As shown in Section 5.2, even when different PEFT methods achieve similar overall accuracy, they tend to produce divergent predictions on individual samples (Figure 1) – this is in accordance with the observations in previous work [A]. Similarly, VFMs pretrained with varying data and objectives demonstrate complementary strengths, with no single model dominating across all examples.
>
> This natural diversity leads to a situation where individual PEFT-VFM pseudo-labels may be biased or inconsistent. By ensembling these diverse predictions, we reduce model-specific noise and extract consensus signals, thereby producing pseudo-labels that are more robust and informative.
>
> Furthermore, we address the practical challenge of inconsistent output scales across models (as discussed in *How to Ensemble?* in section 5.2 and shown in Figure 4) by proposing the *Mean Labels* strategy, which converts predictions into one-hot vectors before averaging. This avoids scale mismatch and ensures a more balanced ensemble.
>
> Together, these designs allow our method to effectively leverage the diversity inherent in VFMs and PEFTs to construct high-quality pseudo-labels. We will take your advice and revise the corresponding section in the paper to make this motivation clearer.
>
>
> **Concern 2: Clarification on Table 3**
>
> We appreciate your feedback and apologize for any confusion. As indicated in the header and the first column of *Table 3*, all baselines and our proposed methods are trained using the same backbone and PEFT method within each part of the table. We will make this clearer in the final revision of the paper.
>
> Regarding your concern about training time, we acknowledge that our ensemble-based methods may introduce additional computational overhead compared to other baselines. However, due to the efficiency of few-shot fine-tuning and pseudo-label extraction, we find the overhead to be _relatively modest considering the performance improvements achieved_. We provide detailed runtime comparisons in the following tables for DTD 3-shot setting, where we estimate the per-epoch wall-clock time as 93 seconds based on our average observation of experiments on one single A100 GPU:
>
> |           | ~Sec/Epoch | Epochs | ~Time (Mins) |
> |-----------|------------|--------|--------------|
> | FixMatch  | 93         | 30     | 46.5         |
> | FlexMatch | 93         | 30     | 46.5         |
> | SoftMatch | 93         | 30     | 46.5         |
> | FineSSL   | 93         | 30     | 46.5         |
>
> |       | ~Sec/Epoch (SFT) | Epochs (SFT) | #SFT Models | ~Sec/Epoch (Ensemble) | Epochs (Ensemble) | ~Total Time (Mins) |
> |-------|------------------|--------------|-------------|------------------------|-------------------|---------------------|
> | PET   | 2                | 50           | 2           | 93                     | 30                | 49.8                |
> | V-PET | 2                | 50           | 4           | 93                     | 30                | 53.1                |
>
> As shown above, PET incurs only ~3.3 additional minutes compared to baselines (1.07×), and V-PET ~6.6 minutes (1.14×). We believe this is a reasonable and practical trade-off considering the performance gains obtained. We will also take your advice and incorporate this table in the revised version of the paper.
>
>
> **Concern 3: Lower Performance on Some Datasets**
>
> Thank you for your careful observation. We acknowledge that although our method consistently outperforms other approaches across the benchmark, its performance gain is less pronounced on certain datasets, namely Resisc45. Diving deep into it, we believe this is due to the nature of the dataset itself. Compared to the rest of the benchmark, Resisc45 is relatively less challenging — as evidenced by the higher performance across other methods in Table 3. In such cases, the outputs of different PEFT models and VFMs tend to be more aligned, leaving limited room for ensembling to further reduce bias or extract complementary information. As a result, the benefit of our ensemble approach is naturally diminished.
>
> However, we would like to respectfully clarify that on **Retinopathy**, our method actually performs competitively — in fact, we outperform most baselines across almost all of the settings, as shown in Table 3. While there may be marginal variations in specific configurations, the overall trend supports the effectiveness of our approach, even on this challenging medical dataset.
>
>
> **Concern 4: Limitations on Scope**
>
> Thank you for your valuable feedback. We acknowledge that extending our work to tasks beyond classification is valuable, but these tasks often require task-specific architecture and training paradigms. To remain focused on core machine learning problems, we align with representative SSL works (e.g., FixMatch) by prioritizing image classification. We humbly believe the fundamental SSL principles and insights from our work are transferable across tasks. For example, UniMatch [B], a popular SSL segmentation method, was inspired by FixMatch. Many SSL segmentation methods [C, D] use self-training & pseudo-labelling. We’ll also take the suggestions and discuss the limitations of our work in the revised version of the paper.
>
> **References**
>
> [A] Lessons and insights from a unifying study of parameter-efficient fine-tuning (PEFT) in visual recognition. CVPR’25.
>
> [B] Revisiting weak-to-strong consistency in semi-supervised semantic segmentation. CVPR’23.
>
> [C] St++: Make self-training work better for semi-supervised semantic segmentation. CVPR’22.
>
> [D] Re-distributing biased pseudo labels for semi-supervised semantic segmentation: A baseline investigation. ICCV21.

---

> > ### Comment · Reviewer_DJbk · 2025-08-05
> >
> > Thank you for the detailed response, which has addressed many of my concerns. However, I have a follow-up question regarding Concern 3.  I agree that the proposed method shows competitive performance overall. However, the explanation that the outputs of PEFT models and VFMs are more aligned on relatively less challenging datasets such as RESISC45—thereby limiting the effectiveness of ensembling—remains somewhat unclear.
> >
> > In particular, on the RESISC45 dataset, while PET consistently improves performance across all cases with the CLIP model, V-PET results in a noticeable drop.

---

> > > ### Author Response · Authors · 2025-08-06
> > >
> > > **Follow-up Response to Concern 3**
> > >
> > > Thank you for your insightful follow-up.
> > >
> > > We agree that RESISC45 exhibits some unique behavior, and we appreciate the opportunity to clarify. In this case, the results appear to be largely driven by the relative quality of pseudo labels (PLs) produced by different backbones.
> > >
> > > Upon closer inspection, CLIP generates substantially better pseudo labels than DINOv2 on RESISC45, as reflected in the higher PL accuracy from CLIP-based models, shown in the table below:
> > >
> > > | Dataset     | Backbone | Method       | PL Accu |
> > > |-------------|----------|--------------|---------|
> > > | Resisc45-N1 | CLIP     | LoRA         | 59.3    |
> > > | Resisc45-N1 | CLIP     | AdaptFormer  | 62.0    |
> > > | Resisc45-N1 | DINOv2   | LoRA         | 48.8    |
> > > | Resisc45-N1 | DINOv2   | AdaptFormer  | 48.7    |
> > > | Resisc45-N2 | CLIP     | LoRA         | 71.2    |
> > > | Resisc45-N2 | CLIP     | AdaptFormer  | 72.5    |
> > > | Resisc45-N2 | DINOv2   | LoRA         | 64.1    |
> > > | Resisc45-N2 | DINOv2   | AdaptFormer  | 56.1    |
> > >
> > > This is contrary to the overall trend, where DINOv2 typically serves as the stronger backbone. As a result:
> > >
> > > - *CLIP + PET* benefits from high-quality self-generated PLs, leading to strong performance.
> > > - *CLIP + V-PET* ensembles PLs from both CLIP and DINOv2. While weaker PLs can sometimes add useful diversity, in this case they slightly degrade the strong CLIP signal.
> > > - *DINOv2 + V-PET*, by contrast, benefits from CLIP’s stronger PLs, leading to a notable improvement over DINOv2 + PET.
> > >
> > > We hope this helps clarify the empirical trend observed on RESISC45.

---

> > > > ### Comment · Reviewer_DJbk · 2025-08-08
> > > >
> > > > Thank you once again for your detailed analysis and clear explanation. Your additional response has fully addressed my remaining concerns. I hope these points will be helpful as you revise the paper.

---

> > > > > ### Author Response · Authors · 2025-08-09
> > > > > **Thank you for the response**
> > > > >
> > > > > Dear Reviewer DJbk,
> > > > >
> > > > > We appreciate your positive feedback. We are glad that our rebuttal has addressed your concerns. We would appreciate it if you would be willing to consider raising your original rating and supporting the acceptance of our paper. Thank you for your consideration.

---

### Official Review · Reviewer_DLdY · 2025-07-02

**Clarity:** 3
**Significance:** 3
**Originality:** 3
**Rating:** 5
**Confidence:** 4

**Summary:**

The authors propose a new pipeline for semi-supervised learning in the regime of pretrained (vision) foundation models (VFMs). They demonstrate several existing SSL methods (which utilize both labelled and unlabelled data for the fine-tuning task) perform worse than a baseline of only using the labelled subset of the data to fine-tune the model on new tasks. The authors then propose their own method, which is more involved but does benefit from unlabelled data, that involves using multiple low-rank fine-tuning methods to fine-tune multiple VFMs to pseudo-label the unlabelled samples and create an ensembled training target.

**Questions:**

See above.

**Ethical Concerns:**

["NO or VERY MINOR ethics concerns only"]

**Final Justification:**

The authors have addressed my concerns:
- All datasets being far-OOD: I later realized that SUN397 is near-OOD. The authors also showed similar results on CUB-200, which is a fine-grained near-OOD dataset.
- Combinations of increasing the number of PEFT methods and VFMs have now been explored sufficiently.
- Computational requirements have been detailed more thoroughly. It transpires that the proposed method is actually less computationally demanding than (most? all? unclear) of the SSL baselines compared against, since the alternative methods rely additional multi-view forward passes of the unlabelled data throughout training. In comparison, the proposed PET method only needs a single additional pass of the unlabelled data through each of the PEFT foundation models before ensembled-target fine-tuning. This difference more than makes up for the additional overhead of the multi-VFM PEFT stage.
- I anticipate formatting issues will be address in revision.

**Limitations:**

Not sufficiently addressed. The proposed method requires more compute than the competitors. The authors do recommend limiting the number of fine-tuned models in the ensemble to 2-4 (L329-330) due to diminishing returns, but do not point out that their approach requires more compute than the competitors since it requires fine-tuning N models on the labelled partition before performing the SSL step.

**Paper Formatting Concerns:**

## Major

Table 1: There is too little white space between the bottom of the table and the start of section 3. Please do not modify margins from the values declared in the stylesheet. I assume the authors have added negative vspace at the bottom of the table. It may be that the formatting issue is accidental; perhaps the authors intended this float to be at the bottom of the page (which would be a better location for it).


## Minor

- L97 has -> have
- L103 "increasingly becoming a common sense"
- L228 "labeled L" -> "labels L" or "labeled dataset L"
- L283 "achieve" -> "achieves"
- Fig 3: It would be good to have this a page sooner to be nearer where the figure is mentioned in the text
- Table 3: The dashed line is visually distracting in this dense table and so I recommend replacing it with a thin (fine) solid line instead
- Fig 5: error bars aren't described - are these different subsets of the methods in the ensemble?
- Fig 6: Please indicate whether higher rank is better or worse - this is sometimes done either way around when showing rankings and so it is ambiguous unless accompanied with text to indicate this.
- L171: Remove blank line
- L298-299: Please cite or otherwise link to each of the VFMs used when they are mentioned here.
- L1012: "Rank Index" -> "Rand Index"
- appendix -> Appendix and section -> Section as these are cased inconsistently in the paper
- Table 8, 10: Text is very large
- Table 12: Text is very small
- Table 12: value 61.91 is misplaced
- References: Some citations have an author and title but no venue. This needs to be included. I suspect this has happened because the papers are arXiv citations with the @misc class. Please change the format of the .bib citations so that the arXiv number is shown in the PDF. Also, peer-reviewed versions of the papers should be preferred over arXiv where available. Also also, citations should ideally have a URL shown in the PDF which makes it easy for a reader to directly go to the cited reference from the paper they are reading.
- References: Some initialisms are not cased correctly (e.g. Sgdr, Dinov2). Please protect these with braces in the .bib file so they render correctly.

**Quality:**

3

**Strengths And Weaknesses:**

## Strengths

The work is well presented and addresses a timely problem. I believe the ideas and implementation are novel. For the most part, I am generally happy with the paper as submitted.

I appreciate that the authors practice a robust experiment protocol with the partition used for hyperparameter optimization distinct from the partition for final model evaluation. I couldn't immediately see details about the partitioning availability so ask the authors to please ensure these splits are made publicly available for reproducibility.


## Weaknesses

**Datasets**: I understand the authors wanting to focus on transfer learning datasets where there is a challenge rather than ones which are trivial to solve given a pretrained VFM out-of-the-box. However, some caution should be taken as the datasets which the authors have selected are relatively far out-of-distribution from the training distribution of these VFMs. Previous work has investigated domain shift and categorized datasets as near- or far-OOD from ImageNet-1k, and the datasets which the authors select are generally on the far side of OOD for image datasets ([Guo, 2019](https://arxiv.org/abs/1912.07200); [Yang, 2022](https://arxiv.org/abs/2210.07242); [Lowe, 2024](https://arxiv.org/abs/2406.02465)). You can tell by eye looking at samples that describable textures (DTD), retinopathy eye scans, Clevr, and satellite imagery (remote sensing) are very different from the natural images that comprise the training datasets usually used for VFMs. For completeness, I think the work would benefit from adding a dataset which is near-OOD. However, being near-OOD does not necessarily mean the dataset isn't challenging - I recommend including a fine-grained classification dataset, which will still be very challenging for the model to learn accurately. There are many datasets used in the fine-grained image classification literature, examples include biodiversity datasets such as [iNaturalist-2021](https://github.com/visipedia/inat_comp/tree/master/2021) and [TreeOfLife-200M](https://huggingface.co/datasets/imageomics/TreeOfLife-200M).

**Experiments**: Unless I missed it, the authors do not seem to explore the interaction between number of VFM models used and number of PEFT methods used in their label-ensembling experiments. It would be helpful to expand the experiments Figure 5 to show the cross between these two to make it clearer when to add more models and when to add more PEFT methods. If I already have used (VFM, PEFT) pair $(f, a)$, would I be better off adding $(g, a)$ a new VFM with an existing PEFT, or $(f, b)$ a new PEFT on an existing VFM, or $(g, b)$ a new PEFT on a new VFM, to the ensemble? Or does it not matter as long as at least one is new?

**Take-away message**: The paper would benefit from giving a take-away message on which VFMs and PEFT methods to prioritize when building the ensemble for pseudo labelling, or a conclusion that it doesn't matter which models/methods are used as long as there is diversity in one/both of these.

**Formatting**: Table 3 is very hard to parse as it is a very large array of numbers. I think this would be easier to read if there were cell background highlighting indicating the magnitude of the values for each column or showing the top K values in each column. This makes the table more like a heatmap so readers can determine trends in which values are large (good) and small (bad). The table would also be easier to read if the redundant % symbol were not shown in every cell. Good practice for table design recommends the units are shown in the first cell and not repeated if the same unit is used in the rest of the row/column. If the same unit is used throughout the table it can instead be described in the caption. Moving the % symbol from the cells to the caption will make the table less busy and hence more legible. Additionally, the font size can be increased with this change.

---

> ### Author Rebuttal · Authors · 2025-07-28
>
> Thank you so much for your interest in our work and your valuable feedback. We would like to address all your concerns, point by point, as follows:
>
> **Concern 1: Dataset Selection**
>
> Thank you for your valuable insight regarding our dataset selection, particularly the observation that our selections are relatively "out-of-distribution (OOD)". We would like to respectively clarify our rationale. We first emphasize that we do not deliberately choose OOD datasets. As our focus is on semi-supervised learning (SSL) in the downstream tasks, we focus on tasks where frozen VFMs underperform, indicating where SSL could offer a beneficial solution [lines 37 - 38]. In retrospect, these datasets turn out to be those that exhibit a larger distribution shift between the pertaining and fine-tuning. That said, with PEFL techniques, the performance gap due to domain shifts can be effectively reduced, suggesting that the pre-training is still beneficial for downstream tasks even under distribution shifts.
>
> Nonetheless, we fully agree with your point of view that incorporating relatively "in-distribution" datasets would create a more representative benchmark. Due to the large scale of [INaturalist-2021](https://github.com/visipedia/inat_comp/tree/master/2021) and [TreeOfLife-200M](https://huggingface.co/datasets/imageomics/TreeOfLife-200M), and limited time during the rebuttal period, we opted to evaluate on [CUB-200](https://huggingface.co/datasets/cassiekang/cub200_dataset) — a smaller-scale, but good quality, and also widely adopted dataset for fine-grained natural image classification. We conducted experiments on it following the same setting as *Table 3* on 4-SHOT task. The results are attached in the table below, with the best performance in each column bolded:
>
> |              | CLIP LoRA  | CLIP AdaptFormer | DINOv2 LoRA | DINOv2 AdaptFormer |
> |--------------|------------|------------------|-------------|--------------------|
> | Labeled Only | 63.95     | 64.65           | 77.30      | 81.03             |
> | FixMatch     | 62.82    | 64.43           | 78.84      | 80.62             |
> | FlexMatch    | 69.62     | 69.21           | 78.05      | 79.06             |
> | SoftMatch    | 71.94     | 71.32           | 81.45      | **82.33**         |
> | FineSSL      | 73.47     | 69.66           | 82.07      | 81.48             |
> | **PET**      | 58.42     | 58.51           | 64.50      | 74.65             |
> | **V-PET**    | **74.61** | **73.68**       | **82.26**  | 82.26             |
>
> Preliminary results on CUB-200 demonstrate that our proposed method also performs well in the fine-grained classification setting. And we’ll leave further exploration to future works.
>
> **Concern 2: Experiment**
>
> We appreciate your thoughtful suggestion regarding the interaction between VFMs and PEFT methods for ensemble methods. We agree that evaluating the interplay between these components is highly valuable and would strengthen our paper.
>
> In response, we conducted the proposed ablation study by varying the number of VFMs (horizontally) and PEFT methods (vertically) from 1 to 6, following the same experimental setup as in Figure 5 of the main paper using DINOv2 and LoRA. The results are presented in an orthogonal grid of configurations, where for each setting, we report the average performance and standard deviation over three randomly sampled combinations.
>
> | N_PEFT \ N_VFM | 1                 | 2                 | 3                 | 4                 | 5                 | 6                 |
> |----------------|-------------------|-------------------|-------------------|-------------------|-------------------|-------------------|
> | 1              | 37.96 ± 9.97    | 61.65 ± 0.79    | 63.99 ± 1.02    | 64.17 ± 0.59    | 63.40 ± 1.02    | 63.40 ± 0.19    |
> | 2              | 61.72 ± 1.30    | 62.68 ± 0.76    | 63.00 ± 0.58    | 62.80 ± 0.67    | 64.45 ± 0.25    | 63.92 ± 0.37    |
> | 3              | 63.23 ± 0.47    | 63.10 ± 0.70    | 63.28 ± 0.76    | 63.81 ± 0.81    | 65.27 ± 0.11    | 64.27 ± 0.46    |
> | 4              | 59.68 ± 3.55    | 63.05 ± 0.84    | 63.86 ± 1.01    | 63.94 ± 0.94    | 65.00 ± 0.16    | 65.12 ± 0.21    |
> | 5              | 61.33 ± 0.04    | 63.40 ± 1.02    | 63.44 ± 0.25    | 63.35 ± 0.87    | 64.11 ± 0.41    | 65.20 ± 0.45    |
> | 6              | 62.48 ± 0.52    | 61.84 ± 0.30    | 63.96 ± 1.24    | 64.22 ± 1.14    | 65.16 ± 0.26    | 65.25 ± 0.16    |
>
> Throughout the table, we can compute the Pearson correlation coefficient between the performance and the number of PEFT (or VFMs), as shown below:
>
> | n_peft | n_vfm |
> |---------------|----------------|
> | 0.272        | 0.446           |
>
> This indicates that the number of VFMs is more related to the overall ensemble performance gain. However, the ensembling over PEFTs method is also useful and can serve as a supplement to VFMs, which is in accordance with the observations made by previous work [A].
>
> **Concern 3: Take-away message for VFMs vs. PEFT**
>
> Thank you for the valuable suggestion. As shown in Figure 5 and our reply to **Concern 2**, the marginal gains from ensembling VFMs consistently exceed those from applying PEFT. Based on this observation, we recommend prioritizing VFMs over PEFT when resources are limited. However, when only a few (or one) VFMs are accessible, PEFT offers an effective alternative to improve performance. We will include this key takeaway message in the revised version of the paper.
>
>
> **Concern 4: Computation Time**
>
> Thank you for your concern. We believe that the computational cost should not be a concern in our work. Like other ensembling-based methods, our approach introduces additional computational overhead. However, thanks to the efficiency of few-shot fine-tuning and pseudo-label generation, the additional cost is actually not severe. Thus, it keeps the overall cost relatively low compared to other ensemble methods.
>
> To quantify this, we report the end-to-end runtime of PET and V-PET compared to standard baselines for DTD 3-shot setting. Based on our average measurements on experiments on an A100 GPU, we estimate the per-epoch time to be approximately 93 seconds.
>
> |           | ~Sec/Epoch | Epochs | ~Time (Mins) |
> |-----------|------------|--------|--------------|
> | FixMatch  | 93         | 30     | 46.5         |
> | FlexMatch | 93         | 30     | 46.5         |
> | SoftMatch | 93         | 30     | 46.5         |
> | FineSSL   | 93         | 30     | 46.5         |
>
> |       | ~Sec/Epoch (SFT) | Epochs (SFT) | #SFT Models | ~Sec/Epoch (Ensemble) | Epochs (Ensemble) | ~Total Time (Mins) |
> |-------|------------------|--------------|-------------|------------------------|-------------------|---------------------|
> | PET   | 2                | 50           | 2           | 93                     | 30                | 49.8                |
> | V-PET | 2                | 50           | 4           | 93                     | 30                | 53.1                |
>
> As shown, PET incurs only ~3.3 additional minutes compared to baselines (1.07×), and V-PET ~6.6 minutes (1.14×). We believe this is a reasonable overhead given the performance gains achieved.
>
> **Concern 5: Formatting issues**
>
> Thank you for pointing out the formatting issues, both about the typos and the table 3 formatting. We will carefully revise the manuscript to correct these problems and ensure a clean, professional presentation in the final version.
>
> **References**
>
> [A] Lessons and insights from a unifying study of parameter-efficient fine-tuning (PEFT) in visual recognition. CVPR’25.

---

> ### Comment · Reviewer_DLdY · 2025-08-04
>
> Thank you for the response. I have some clarifying follow-up questions.
>
> **Concern 1: Dataset Selection**
>
> Looking at the datasets in the paper again, I realize that it includes one which is near-OOD: SUN397. This mitigates some of my previous concern that all the datasets were far-OOD from ImageNet.
>
> I thank the authors for running these additional experiments, demonstrating similar trends for CUB-200 as seen for the other datasets. (The TreeOfLife dataset was a bad suggestion, not just because of its size but also because it includes data from both lab and field settings, meaning the distribution is inconsistent.) I have a question for the authors: do they intend to add the CUB-200 results to the paper?
>
> **Concern 2: Experiment**
>
> Thank you for sharing these additional results. Can you clarify - does the number of fine-tuned models used for the ensemble equal the product of N_PEFT and N_VFM? In that case, looking at the iso-compute groups, it does indeed seem that adding more VFMs dominates over adding more PEFTs, e.g. comparing N_PEFT * N_VFM = 6:
> - N_PEFT=1, N_VFM=6 => 63.40 ± 0.19
> - N_PEFT=2, N_VFM=3 => 63.00 ± 0.58
> - N_PEFT=3, N_VFM=2 => 63.10 ± 0.70
> - N_PEFT=6, N_VFM=1 => 62.48 ± 0.52
>
> Note however, these still leaves open the possibility of using a mixture of different PEFT methods for each VFM (as I suggested in my review). Using one PEFT method for each of the 6 VFMs, but choosing a different PEFT method for each, could be more beneficial than the N_PEFT=1, N_VFM=6 setting, without imposing additional computational costs, and may be worth considering.
>
> **Concern 3: Take-away message for VFMs vs. PEFT**
>
> I am happy with this conclusion, and support its addition to strengthen the paper.
>
> **Concern 4: Computation Time**
>
> Thank you for the clarification regarding the computational requirements. I am surprised that the costs of the PEFT stage is so light per model. Since three of the reviewers highlighted this as a concern, I think it would be prudent to update the paper to specify the computational requirements of each of the methods considered in Table 3. This would improve the impact of the paper, since people are more likely to use the method if they correctly understand the additional computational overhead is light.

---

> ### Author Response · Authors · 2025-08-05
>
> **Reply for Concern 1**
>
> Thank you for your recognition of our dataset selection, the additional experiments, and your continued interest.
>
> Regarding the CUB-200 results, we are currently not planning to include them in the main table. The main reason is that our benchmark strictly follows a subset of VTAB [A], where we select exactly two datasets from each of VTAB’s official categories:
>
> (1) *Natural* – DTD, SUN397
> (2) *Specialized* – RESISC45, DIABETIC-RETINOPATHY
> (3) *Structured* – CLEVR-COUNT, KITTI
>
> Introducing a new dataset from outside would disrupt this structure. We will ensure this methodology is clearly reiterated in the revised version. That said, we are happy to include the CUB-200 results in the additional experimental sections.
>
> **Reply for Concern 2**
>
> Thank you for bringing up your further concerns. As for your question about the number of fine-tuned models — yes, the number of ensembled pseudo-labels equals *N_PEFT × N_VFM*. We also appreciate your sharp observation that, under equal computation, *N_VFM* scales better than *N_PEFT*. This provides additional empirical support beyond the Pearson correlation coefficient.
>
> We will include a discussion in the revised version summarizing this exchange — including your suggestion of using a mixture of different PEFT methods across VFMs (e.g., assigning a different PEFT method to each of the 6 VFMs in the *N_PEFT*=1, *N_VFM*=6 setting), as well as the additional experiments we conducted during the rebuttal process.
>
> **Reply for Concerns 3 & 4**
>
> Thank you for your positive feedback regarding our additional results. We also appreciate your suggestion to clarify the computational requirements. As you pointed out, this is a common concern raised by several reviewers. We will include the corresponding compute estimates and conclusions in the revised version.
>
> We hope the clarifications and additional results have addressed your concerns, and would appreciate your consideration in reflecting this in your final score.
>
> ---
> **References**
>
> [A] A Large-scale Study of Representation Learning with the Visual Task Adaptation Benchmark, 2019

---

> > ### Comment · Reviewer_DLdY · 2025-08-05
> >
> > As my concerns have been addressed satisfactorily, I have raised my score accordingly, and thank the authors for their response.

---

> > > ### Author Response · Authors · 2025-08-07
> > > **Thank you for your positive feedback**
> > >
> > > Dear Reviewer DLdY,
> > >
> > > We are glad that our rebuttal has satisfactorily addressed your concern, and we sincerely appreciate your willingness to raise your score (original score: 4: Borderline accept).
> > >
> > > For your information, we noticed that **the score in the system has stayed unchanged**, and we look forward to the updated score and justification.
> > >
> > > Once again, thank you for your thoughtful comments and your positive recognition of our work.

---

> > ### Comment · Reviewer_DLdY · 2025-08-08
> >
> > As reviewer gEWg pointed out, there is another computational cost which the authors neglected to include when reporting the additional computational overhead to the reviewers.
> >
> > Assuming the forward pass is around a third of the time for a training step, then running step 2 of the training pipeline for V-PET in which the unlabelled data is passed through each of the 4 PEFT fine-tuned models costs around 4/3*93 seconds = 2.1 mins.
> >
> > I ask the authors to please include the full computation time of their method when updating the paper to include the time costs.
> >
> > By my estimation, including the extra forward-pass of the unlabelled data through each of the PEFT models brings the total cost for V-PET to at least 8.73 mins, which is 1.19x as long as the compute time for the baseline competitors. This then begs the question - how much better would the baseline methods do if they could take advantage of this increase in the compute budget and train for ~36 epochs instead of 30? Or (a simpler way of doing a comparison in terms of equal compute time) what if V-PET only gets a total runtime of 46.5 mins to match the others, meaning its ensemble-guided training step only gets 24.4 epochs instead of 30? When it is a 20% difference in compute budget for the methods, I do not think the comparison is sufficiently fair.

---

> ### Author Response · Authors · 2025-08-09
>
> **Response: Regarding the Additional Forward Pass Cost**
>
> We appreciate the reviewer for cross-checking other reviewers’ questions and our responses. We acknowledge that we neglected the forward pass of multiple VFMs in generating pseudo-labels, and we apologize for it. In our new response to reviewer **gEWg** (on August 8th, 2025), we bring back such a computational overhead.
>
> We note that the forward pass in which the unlabelled data is passed through each of the 4 PEFT fine-tuned models costs **less** than 4/3*93 seconds = 2.1 minutes. This is because during pure forward passes (without backward passes), a larger batch size can be used compared to the one used in training. Based on the logs, the time for each VFM to forward pass through all unlabeled data is *~15 seconds*, which becomes *~60 seconds* when considering four models. In our original rebuttal, we found this time smaller than a single epoch and ignored it, but we agree with you that it would be better to report it for a more accurate report of our computational time.
>
> **Response: Full Computation Wall-Clock Time**
>
> We appreciate your suggestion to report the full, wall-clock computation time. The reason why we did not report it in the original rebuttal because we found some implementation details in the baselines (e.g., FixMatch, FlexMatch, and SoftMatch) may inadvertently introduce additional computation. For example, each algorithm extracts logits for multiple augmentations (e.g., no, weak, or strong), requiring more than a single forward pass through each training sample (in other words, a single training sample is seen more than once per epoch). Indeed, if we report the end-to-end wall-clock time without modifying the baselines, our method runs **slightly faster**.
>
> | Method  | Total Time (Mins) |
> |--------|------------------|
> | FixMatch    | 35.82         |
> | FlexMatch  | 35.56         |
> | SoftMatch  | 36.61         |
>
> | Method | Total Time (Mins) |
> |--------|------------------|
> | PET    | 29.57              |
> | V-PET | 32.12              |
>
> *NOTE: We rerun the experiments on a supercomputer center, obtaining absolute times slightly faster than those reported earlier in the rebuttal, which were based on the logs from the paper submission period. We hypothesize that this may be caused by an infrastructure upgrade, which should not affect the relative time across methods.*
>
> **That said, these results certainly do not mean our method is more computationally efficient.** We report them here mainly to justify why we did not report the real wall-clock time in the first place, but rather report an estimated time — **we aim to report a time that reflects the difference introduced by the use of VFMs, rather than training details**. We hope this clarification addresses your concern.
>
> **Response: Regarding Computation Time and the Number of Epochs**
>
> We appreciate your calculation regarding the additional time our methods need and whether it could be traded for more training epochs for the baselines. In our experience, 30 epochs are sufficient for the baselines and our methods to converge; reducing 10%~20% of training epochs introduces a negligible drop to our approach. In our humble opinion, considering the number of epochs would bring in an extra factor—the convergence speed of different methods—into consideration, which may defocus the comparison in this paper.
>
> If we understand your and reviewer **gEWg**’s original review correctly, the concern was whether our method would introduce a **significant linear scaling** in terms of computation time, as we bring in multiple VFMs. In this regard, we respectfully believe that a **1.16x** increase demonstrates that our method does not result in a significant overhead. This is because the computational overhead of multiple VFMs only appears in stages 1-3, while stage 4 remains the major computational bottleneck (based on our new response to reviewer **gEWg** on August 8th). Indeed, if we consider the experiments in earlier rebuttal post, where we studied 6 VFMs*6 PEFT methods, the overall computational time would go to ~115.5 minutes, roughly 2.48x over baselines instead of 36x.
>
> We believe that a **1.16x** difference in computational time is acceptable, not to mention that the implementation details may indeed introduce additional overheads for the baselines.
>
> We hope our further clarifications above address your concerns, and we will be happy to provide further discussions.

---

### Decision · Program_Chairs · 2025-09-17

**Decision:**

Accept (poster)

**Comment:**

This paper revisits semi-supervised learning (SSL) in the context of vision foundation models (VFMs). Through a systematic study, the authors show: (1) parameter-efficient fine-tuning (PEFT) on labeled data alone can outperform classic SSL baselines; (2) PEFT models provide high-quality pseudo-labels; and (3) complementary signals emerge when ensembling pseudo-labels across diverse PEFT methods and VFM backbones. They propose a simple baseline—ensemble pseudo-labeling across heterogeneous PEFT techniques and VFMs—and validate it extensively.

All reviewers recommend acceptance, highlighting strong empirical results, a clear and timely problem framing, and actionable takeaways for practitioners adapting SSL to the “foundation model era.” The paper offers clear, empirically grounded insights that reshape how we think about SSL in the presence of strong VFMs and PEFT. The proposed ensemble pseudo-labeling baseline is simple and likely to influence practice. Please address the notes made by the reviewers to strengthen the camera-ready version.